# HippoUnit: A software tool for the automated testing and systematic comparison of detailed models of hippocampal neurons based on electrophysiological data

Sára Sáray[1,2]*, Christian A. Rössert[3], Shailesh Appukuttan[4], Rosanna Migliore[5], Paola Vitale[5], Carmen A. Lupascu[5], Luca L. Bologna[5], Werner Van Geit[3], Armando Romani[3], Andrew P. Davison[4], Eilif Muller[3,6,7,8], Tamás F. Freund[1,2], Szabolcs Káli[1,2]*

1 Faculty of Information Technology and Bionics, Pázmány Péter Catholic University, Budapest, Hungary, 2 Institute of Experimental Medicine, Budapest, Hungary, 3 Blue Brain Project, École Polytechnique Fédérale de Lausanne, Geneva, Switzerland, 4 Paris-Saclay Institute of Neuroscience, Centre National de la Recherche Scientifique/Université Paris-Saclay, Gif-sur-Yvette, France, 5 Institute of Biophysics, National Research Council, Palermo, Italy, 6 Department of Neurosciences, Faculty of Medicine, University of Montreal, Montreal, Canada, 7 CHU Sainte-Justine Research Center, Montreal, Canada, 8 Quebec Artificial Intelligence Institute (Mila), Montreal, Canada

* saray.sara@koki.hu (SS); kali@koki.hu (SK)

**Data Availability Statement:** The versions of the models from literature that were tested in this

## Abstract

Anatomically and biophysically detailed data-driven neuronal models have become widely used tools for understanding and predicting the behavior and function of neurons. Due to the increasing availability of experimental data from anatomical and electrophysiological measurements as well as the growing number of computational and software tools that enable accurate neuronal modeling, there are now a large number of different models of many cell types available in the literature. These models were usually built to capture a few important or interesting properties of the given neuron type, and it is often unknown how they would behave outside their original context. In addition, there is currently no simple way of quantitatively comparing different models regarding how closely they match specific experimental observations. This limits the evaluation, re-use and further development of the existing models. Further, the development of new models could also be significantly facilitated by the ability to rapidly test the behavior of model candidates against the relevant collection of experimental data. We address these problems for the representative case of the CA1 pyramidal cell of the rat hippocampus by developing an open-source Python test suite, which makes it possible to automatically and systematically test multiple properties of models by making quantitative comparisons between the models and electrophysiological data. The tests cover various aspects of somatic behavior, and signal propagation and integration in apical dendrites. To demonstrate the utility of our approach, we applied our tests to compare the behavior of several different rat hippocampal CA1 pyramidal cell models from the ModelDB database against electrophysiological data available in the literature, and evaluated how well these models match experimental observations in different domains. We also show how we employed the test suite to aid the development of models within the European

study, along with the target experimental data, the validation results and the Jupyter notebooks that can be used to reproduce the results, are available in a GitHub repository: https://github.com/KaliLab/HippoUnit_demo. All the validation tests, the tested models and the validation results presented here are registered in the Validation Framework of the Human Brain Project and are available via the Live Paper: https://humanbrainproject.github.io/hbp-bsp-live-papers/2020/saray_et_al_2020/saray_et_al_2020.html. Access to this resource requires free user registration at https://www.humanbrainproject.eu/en/hbp-platforms/getting-access/.

**Funding:** This project received funding from the European Union's Horizon 2020 Framework Programme for Research and Innovation under Specific Grant Agreements No. 720270 and No. 785907 (Human Brain Project SGA1 and SGA2). SS has been supported by the ÚNKP-19-3-III New National Excellence Program of the Ministry For Innovation and Technology (Hungary), and the European Union, co-financed by the European Social Fund (EFOP-3.6.3-VEKOP- 16-2017-00002). The funders had no role in study design, data collection and analysis, decision to publish, or preparation of the manuscript.

**Competing interests:** The authors have declared that no competing interests exist.

Human Brain Project (HBP), and describe the integration of the tests into the validation framework developed in the HBP, with the aim of facilitating more reproducible and transparent model building in the neuroscience community.

## Author summary

Anatomically and biophysically detailed neuronal models are useful tools in neuroscience because they allow the prediction of the behavior and the function of the studied cell type under circumstances that are hard to investigate experimentally. However, most detailed biophysical models have been built to capture a few selected properties of the real neuron, and it is often unknown how they would behave under different circumstances, or whether they can be used to successfully answer different scientific questions. To help the modeling community develop better neural models, and to make the process of model building more reproducible and transparent, we developed a test suite that enables the comparison of the behavior of models of neurons in the rat hippocampus and their evaluation against experimental data. Applying our tests to several models available in the literature enabled us to assess and compare how precisely each of these models is able to match various electrophysiological properties of the real neurons. We also use the test suite in the model development workflow of the European Human Brain Project to aid the construction of better models of hippocampal neurons and networks.

## Introduction

The construction and simulation of anatomically and biophysically detailed models is becoming a standard tool in neuroscience [1]. Such models, which typically employ the compartmental modeling approach and a Hodgkin-Huxley-type description of voltage-gated ion channels, are capable of providing fairly accurate models of single neurons [2–10] and (when complemented by appropriate models of synaptic interactions) even large-scale circuits [11–14]. However, building such detailed multi-compartmental models of neurons requires setting a large number of parameters (such as the densities of various ion channels in multiple neuronal compartments) that are often not directly constrained by the available experimental data. These parameters are typically tuned (either manually or using automated parameter-search methods [9,15–17]) until the simulated physiological behavior of the model matches some pre-defined set of experimental observations.

For an increasing number of cell types, the available experimental data already provide diverse constraints on the expected physiological behavior of the neuron under a variety of conditions. Based on various (typically small) subsets of the available constraints, a large number of different models of several cell types have been developed to investigate diverse aspects of single-cell behavior, and for inclusion in realistic circuit models. As an example, there are currently 131 different models related to the hippocampal CA1 pyramidal cell (PC) in the ModelDB database [18]. However, even though these models are publicly available, it is still technically challenging to verify their behavior beyond the examples explicitly included with the model, and especially to test their behavior outside the context of the original study, or to compare it with the behavior of other models. This sparsity of information about the performance of detailed models may also be one reason why model re-use in the community is relatively limited, which decreases the chance of spotting errors in modeling studies, and may lead to an unnecessary replication of effort.

A systematic comparison of existing models built in different laboratories requires the development of a comprehensive validation suite, a set of automated tests that quantitatively compare various aspects of model behavior with the corresponding experimental data. Such validation suites enable all modeling groups to evaluate their existing and newly developed models according to the same set of well-defined criteria, thus facilitating model comparison and providing an objective measure of progress in matching relevant experimental observations. Applying automated tests also allows researchers to learn more about models published by other groups (beyond the results included in the papers) with relatively little effort, thus facilitating optimal model re-use and co-operative model development. In addition, systematic, automated testing is expected to speed up model development in general by allowing researchers to easily evaluate models in relation to the relevant experimental data after every iteration of model adjustment. Finally, a comprehensive evaluation of model behavior appears to be critical for models that are then expected to provide useful predictions in a new context. A prime example of this is detailed single cell models included in network models, where diverse aspects of cellular function such as synaptic integration, intracellular signal propagation, spike generation and adaptation mechanisms all contribute to the input-output function of the neuron in the context of an active network. By comparing multiple different aspects of the behavior of the single cell model with experimental data, one can increase the chance of having a model that also behaves correctly within the network. The technical framework for developing automated test suites for models already exists [19], and is currently used by several groups to create a variety of tests for models of neural structure and function at different scales [20–24]. In the current study, our goal was to develop a validation suite for the physiological behavior of one of the most studied cell types of the mammalian brain, the pyramidal cell in area CA1 of the rat hippocampus.

CA1 pyramidal neurons display a large repertoire of nonlinear responses in all of their compartments (including the soma, axon, and various functionally distinct parts of the dendritic tree), which are experimentally well-characterized. In particular, there are detailed quantitative results available on the subthreshold and spiking voltage response to somatic current injections [3,25]; on the properties of the action potentials back-propagating from the soma into the dendrites [26–28], which is a basic measure of dendritic excitability; and on the characteristics of the spread [29] and non-linear integration of synaptically evoked signals in the dendrites, including the conditions necessary for the generation of dendritic spikes [30–33].

The test suite that we have developed allows the quantitative comparison of the behavior of anatomically and biophysically detailed models of rat CA1 pyramidal neurons with experimental data in all of these domains. In this paper, we first describe the implementation of the HippoUnit validation suite. Next, we show how we used this test suite to systematically compare existing models from six prominent publications from different laboratories. We then show an example of how the tests have been applied to aid the development of new models in the context of the European Human Brain Project (HBP). Finally, we describe the integration of our test suite into the general validation framework developed in the HBP.

## Methods

### Implementation of HippoUnit

HippoUnit is a Python test suite based on the SciUnit [19] framework, which is a Python package for testing scientific models, and during its implementation the NeuronUnit package [20] was taken into account as an example of how to use the SciUnit framework for testing neuronal models. In SciUnit tests usually four main classes are implemented: the test class, the model class, the capabilities class and the score class. HippoUnit is built in a way that keeps

this structure. The key idea behind this structure is the decoupling of the model implementation from the test implementation by defining standardized interfaces (capabilities) between them, so that tests can easily be used with different models without being rewritten, and models can easily be adapted to fit the framework.

Each test of HippoUnit is a separate Python class that, similarly to other SciUnit packages, can run simulations on the models to generate model *predictions*, which can be compared with experimental *observations* to yield the final score, provided that the model has the required capabilities implemented to mimic the appropriate experimental protocol and produce the same type of measurable output. All measured or calculated data that contribute to the final score (including the recorded voltage traces, the extracted features and the calculated feature scores) are saved in JSON or pickle files (or, in many cases, in both types of files). JSON files are human readable, and can be easily loaded into Python dictionaries. Data with a more complex structure are saved into pickle files. This makes it possible to easily write and read the data (for further processing or analysis) without changing its Python structure, no matter what type of object or variable it is. In addition to the JSON files a text file (log file) is also saved, that contains the final score and some useful information or notes specific to the given test and model. Furthermore, the recorded voltage traces, the extracted features and the calculated feature scores are also plotted for visualization.

Similarly to many of the existing SciUnit packages the implementations of specific models are not part of the HippoUnit package itself. Instead, HippoUnit contains a general `Model-Loader` class. This class is implemented in a way that it is able to load and deal with most types of models defined in the HOC language of the NEURON simulator (either as standalone HOC models or as HOC templates) [34]. It implements all model-related methods (capabilities) that are needed to simulate these kinds of neural models in order to generate the prediction without any further coding required from the user.

For the smooth validation of the models developed using parameter optimization within the HBP there is a child class of the `ModelLoader` available in HippoUnit that is called `ModelLoader_BPO`. This class inherits most of the functions (especially the capability functions) from the `ModelLoader` class, but it implements additional functions that are able to automatically deal with the specific way in which information is represented and stored in these optimized models. The role of these functions is to gather all the information from the metadata and configuration files of the models that are needed to set the parameters required to load the models and run the simulations on them (such as path to the model files, name of the model template or the simulation temperature (the `celsius` variable of Neuron)). This enables the validation of these models without any manual intervention needed from the user. The section lists required by the tests of HippoUnit are also created automatically using the morphology files of these models (for details see the "Classify apical sections of pyramidal cells" subsection). For neural models developed using other software and methods, the user needs to implement the capabilities through which the tests of HippoUnit perform the simulations and recordings on the model.

The capabilities are the interface between the tests and the models. The `ModelLoader` class inherits from the capabilities and must implement the methods of the capability. The test can only be run on a model if the necessary capability methods are implemented in the `ModelLoader`. All communication between the test and the model happens through the capabilities.

The methods of the score classes perform the quantitative comparison between the *prediction* and the *observation*, and return the score object containing the final score and some related data, such as the paths to the saved figure and data (JSON) files and the prediction and observation data. Although SciUnit and NeuronUnit have a number of different score types

implemented, those typically compare a single *prediction* value to a single *observation* value, while the tests of HippoUnit typically extract several features from the model's response to be compared with experimental data. Therefore, each test of HippoUnit has its own score class implemented that is designed to deal with the specific structure of the output *prediction* data and the corresponding *observation* data. For simplicity, we refer to the discrepancy between the target experimental data (*observation*) and the models' behavior (*prediction*) with respect to a studied feature using the term feature score. In most cases, when the basic statistics (mean and standard deviation) of the experimental features (typically measured in several different cells of the same cell type) are available, feature scores are computed as the absolute difference between the feature value of the model and the experimental mean feature value, divided by the experimental standard deviation (Z-score) [35]. The final score of a given test achieved by a given model is given by the average (or, in some cases, the sum) of the feature scores for all the features evaluated by the test.

## Implementation of the tests of HippoUnit

**The Somatic Features Test.**   The Somatic Features Test uses the Electrophys Feature Extraction Library (eFEL) [36] to extract and evaluate the values of both subthreshold and suprathreshold (spiking) features from voltage traces that represent the response of the model to somatic current injections of different positive (depolarizing) and negative (hyperpolarizing) current amplitudes. Spiking features describe action potential shape (such as AP width, AP rise/fall rate, AP amplitude, etc.) and timing (frequency, inter-spike intervals, time to first/last spike, etc.), while some passive features (such as the voltage base or the steady state voltage), and subthreshold features for negative current stimuli (voltage deflection, sag amplitude, etc.) are also examined.

In this test step currents of varying amplitudes are injected into the soma of the model and the voltage response is recorded. The simulation protocol is set according to an input configuration JSON file, which contains all the current amplitudes, the delay and the duration of the stimuli, and the stimulation and recording positions. Simulations using different current amplitudes are run in parallel if this is supported by the computing environment.

As the voltage responses of neurons to somatic current injections can strongly depend on the experimental method, and especially on the type of electrode used, target values for these features were extracted from two different datasets. One dataset was obtained from sharp electrode recordings from adult rat CA1 neurons (this will be called the sharp electrode dataset) [3], and the other dataset is from patch clamp recordings in rat CA1 pyramidal cells (data provided by Judit Makara, which will be referred to as the patch clamp dataset). For both of these datasets we had access to the recorded voltage traces from multiple neurons, which made it possible to perform our own feature extraction using eFEL. This ensures that the features are interpreted and calculated the same way for both the experimental data and the models' voltage response during the simulation. Furthermore, it allows a more thorough comparison against a large number of features extracted from experimental recordings yielded using the exact same protocol, which is unlikely to be found in any paper of the available literature. However, to see how representative these datasets are of the literature as a whole we first compared some of the features extracted from these datasets to data available on Neuroelectro.org [37] and on Hippocampome.org [38]. The features we compared were the following: resting potential, voltage threshold, after-hyperpolarization (AHP) amplitudes (fast, slow), action potential width and sag ratio. Although these databases have mean and standard deviation values for these features that are calculated from measurements using different methods, protocols and from different animals, we found that most of the feature values for our two

experimental datasets fall into the ranges declared as typical for CA1 PCs in the online databases. The only conspicuous exception is the fast AHP amplitude of the patch clamp dataset used in this study, which is 1.7 ± 1.5 mV, while the databases cite values between 6.8 and 11.64 mV. This deviation could possibly stem from a difference in the way that the fast AHP is measured. However, we note that during the patch clamp recordings some of the cells were filled with a high-affinity $Ca^{2+}$ sensor, which may have affected several Ca-sensitive mechanisms (such as Ca-dependent potassium currents) in the cell, and therefore may have influenced features like the AP width and properties of the spike after-hyperpolarization.

We also performed a more specific review of the relevant literature to compare the most important somatic features of the patch clamp dataset to results from available patch clamp recordings (Table 1). Our analysis confirmed that the values of several basic electrophysiological features such as the AP voltage threshold, the AP amplitude, the AP width, and the amplitude of the hyperpolarizing sag extracted from our patch clamp dataset fall into the range observed experimentally. We conclude that the patch clamp dataset is in good agreement with experimental observations available in the literature, and will be used as a representative example in this study.

The *observation* data are loaded from a JSON file of a given format which contains the names of the features to be evaluated, the current amplitude for which the given feature is evaluated and the corresponding experimental mean and standard deviation values. The feature means and standard deviations are extracted using BluePyEfe [45] from a number of voltage traces recorded from several different cells. Its output can be converted to stimulus and feature JSON files used by HippoUnit using the script available here: https://github.com/sasaray/HippoUnit_demo/blob/master/target_features/Examples_on_creating_JSON_files/Somatic_Features/convert_new_output_feature_data_for_valid.py. Setting the `specify_data_set` parameter it can be ensured that the test results against different experimental data sets are saved into different folders.

For certain features eFEL returns a vector as a result; in these cases, the feature value used by HippoUnit is the average of the elements of the vector. These are typically spiking features for which eFEL extracts a value corresponding to each spike fired. For features that use the 'AP_begin_time' or 'AP_begin_voltage' feature values for further calculations, we exclude the first element of the vector output before averaging because we discovered that these features are often incorrectly detected for the first action potential of a train.

**Table 1. Comparison of the most important somatic features extracted using eFEL from the patch clamp dataset (used as target data in the Somatic Features Test) to results from patch clamp recordings available in the literature.**

| Feature<br>(eFEL feature name) | Value in literature | Value in patch clamp dataset |
|---|---|---|
| AP voltage threshold<br>(AP_begin_voltage) | -46 - -53 mV [39–42] | -51.13±0.97 mV (0.15 nA current step)<br>-50.14±1.97 mV (0.2 nA current step)<br>-49.36±2.02 mV (0.25 nA current step) |
| AP amplitude<br>(AP_amplitude_from_voltagebase) | 71–112 mV [39,42,43] | 98.36±5.82 mV (0.15 nA current step)<br>96.83±5.66 mV (0.2 nA current step)<br>95.99±5.22 mV (0.25 nA current step) |
| AP width at half amplitude<br>(AP_duration_half_width) | 0.8–1.29 ms [39,41–43] | 1.23±0.096 ms (0.15 nA current step)<br>1.25±0.11 ms (0.2 nA current step)<br>1.32±0.086 ms (0.25 nA current step) |
| sag ratio<br>(sag_ratio2) | 0.84±0.02 [43],<br>0.83±0.01 [44] | 0.79±0.023 (-0.05 nA current step)<br>0.81±0.03 (-0.1 nA current step)<br>0.81±0.027 (-0.15 nA current step)<br>0.81±0.03 (-0.2 nA current step)<br>0.80±0.03 (-0.25 nA current step) |

The score class of this test returns as the final score the average of *Z-scores* for the evaluated eFEL features achieved by the model. Those features that could not be evaluated (e.g., spiking features from voltage responses without any spikes) are listed in a log file to inform the user, and the number of successfully evaluated features out of the number of features attempted to be evaluated is also reported.

**The Depolarization Block Test.**   This test aims to determine whether the model enters depolarization block in response to a prolonged, high intensity somatic current stimulus. For CA1 pyramidal cells, the test relies on experimental data from Bianchi et al. [25]. According to these data, rat CA1 PCs respond to somatic current injections of increasing intensity with an increasing number of action potentials until a certain threshold current intensity is reached. For current intensities higher than the threshold, the cell does not fire over the whole period of the stimulus; instead, firing stops after some action potentials, and the membrane potential is sustained at some constant depolarized level for the rest of the stimulus. This phenomenon is termed depolarization block [25].

This test uses the same capability class as the Somatic Features Test for injecting current and recording the somatic membrane potential (see the description above). Using this capability, the model is stimulated with 1000 ms long square current pulses increasing in amplitude from 0 to 1.6 nA in 0.05 nA steps, analogous to the experimental protocol. The stimuli of different amplitudes are run in parallel. Somatic spikes are detected and counted using eFEL [36].

From the somatic voltage responses of the model, the following features are evaluated. $I_{th}$ is the threshold current to reach depolarization block; experimentally, this is both the amplitude of the current injection at which the cell exhibits the maximum number of spikes, and the highest stimulus amplitude that does not elicit depolarization block. In the test two separate features are evaluated for the model and compared to the experimental $I_{th}$: the current intensity for which the model fires the maximum number of action potentials (*I_maxNumAP*), and the current intensity one step before the model enters depolarization block (*I_below_depol_block*). If these two feature values are not equal, a penalty is added to the score. The model is defined to exhibit depolarization block if *I_maxNumAP* is not the highest amplitude tested, and if there exists a current intensity higher than *I_maxNumAP*, for which the model does not fire action potentials during the last 100 ms of its voltage response.

In the experiment the $V_{eq}$ feature is extracted from the voltage response of the pyramidal cells to the current injection one step above $I_{th}$ (or *I_max_num_AP* in the test). Both in the experiment and in this test this is calculated as the mean voltage over the last 100 ms of the voltage trace. However, in the test, before calculating this value it is examined whether there are any action potentials during this period. The presence of spikes here means that the model did not enter depolarization block prior to this period. In these cases the test iterates further on the voltage traces corresponding to larger current steps to find if there is any where the model actually entered depolarization block; if an appropriate trace is found, the value of $V_{eq}$ is extracted there. This trace is the response to the current intensity one step above *I_below_depol_block*.

If the model does not enter depolarization block, a penalty is applied, and the final score gets the value of 100. Otherwise, the final score achieved by the model on this test is the average of the feature scores (Z-scores) for the features described above, plus an additional penalty if *I_maxNumAP* and *I_below_depol_block* differ. This penalty is 200 times the difference between the two current amplitude values (in pA–which in this case is 10 times the number of examined steps between them).

**The Back-propagating AP Test.**   This test evaluates the strength of action potential back-propagation in the apical trunk at locations of different distances from the soma. The observation data for this test were yielded by the digitization of Fig 1B of [27], using the DigitizeIt

software [46]. The values were then averaged over distances of 50, 150, 250, 350 ± 20 μm from the soma to get the mean and standard deviation of the features. The features tested here are the amplitudes of the first and last action potentials of a 15 Hz spike train, measured at the 4 different dendritic locations.

The test automatically finds current amplitudes for which the soma fires, on average, between 10–20 Hz and chooses the amplitude that leads to firing nearest to 15 Hz. For this task, the following algorithm was implemented. Increasing current step stimuli of 0.0–1.0 nA amplitude with a step size of 0.1 nA are applied to the model and the number of spikes is counted for each resulting voltage trace. If spontaneous spiking occurs (i.e., if there are spikes even when no current is injected) or if the spiking rate does not reach 10 Hz even for the highest amplitude, the test quits with an error message. Otherwise the amplitudes for which the soma fires between 10 and 20 Hz are appended to a list and (if the list is not empty) the one providing the spiking rate nearest to 15 Hz is chosen. If the list is empty because the spiking rate is smaller than 10 Hz for a step amplitude but higher than 20 Hz for the next step, a binary search method is used to find an appropriate amplitude in this range.

This test uses a trunk section list (or generates one if the `find_section_lists` variable of the `ModelLoader` is set to True–see the section 'Classifying the apical sections of pyramidal cells' below) to automatically find the dendritic locations for the measurements. The desired distances of the locations from the soma and the distance tolerance are read from the input configuration file, and must agree with the distances and the tolerance over which the experimental data were averaged. All the trunk dendritic segments whose distance from the soma falls into one of the distance ranges are selected. The locations and also their distances are then returned in separate dictionaries.

Then the soma is stimulated with a current injection of the previously chosen amplitude and the voltage response of the soma and the selected dendritic locations are recorded and returned.

The test implements its own function to extract the amplitudes of back-propagating action potentials, but the method is based on eFEL features. This is needed because eFEL's spike detection is based on a given threshold value for spike initiation, which may not be reached by the back-propagating signal at more distant regions. First the maximum depolarization of the first and the last action potentials are calculated. This is the maximum value of the voltage trace in a time interval around the somatic action potential, based on the start time of the spike (using the AP_begin_time feature of eFEL) and the inter-spike interval to the next spike recorded at the soma. Then the amplitudes are calculated as the difference between this maximum value and the voltage at the begin time of the spike (on the soma) minus 1 ms (which is early enough not to include the rising phase of the spike, and late enough in the case of the last action potential not to include the afterhyperpolarization of the previous spike).

To calculate the feature scores the amplitude values are first averaged over the distance ranges to be compared to the experimental data and get the feature Z-scores. The final score here is the average of the Z-scores achieved for the features of first and last action potential amplitudes at different dendritic distances. In the result it is also stated whether the model is more like a strongly or a weakly propagating cell in the experiment, where they found examples of both types [27].

**The PSP Attenuation Test.** The PSP Attenuation Test evaluates how much the post-synaptic potential attenuates as it propagates from different dendritic locations to the soma in rat CA1 pyramidal cell models. The *observation* data for this test were yielded by the digitization of Fig 1E and Fig 2B of Magee and Cook, 2000 [29] using the DigitizeIt software [46]. The somatic and dendritic depolarization values were then averaged over distances of 100, 200, 300 ± 50 μm from the soma and the soma/dendrite attenuation was calculated to get the mean

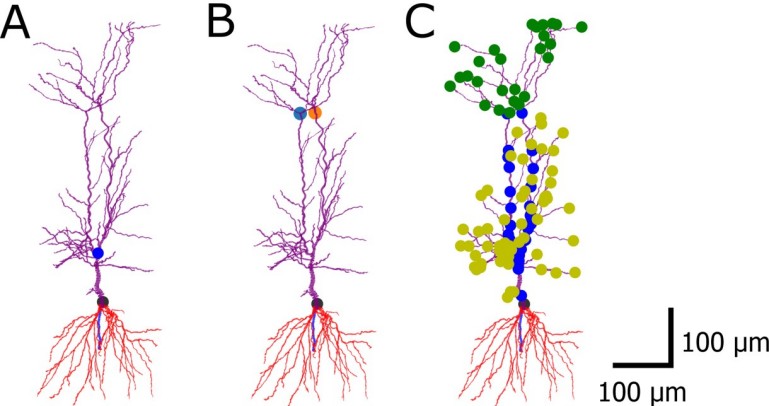

**Fig 1. Classifying the apical dendrites of pyramidal cells.** Morphological reconstruction made within the HBP at University College London (UCL). The soma is marked in black, the red dendrites underneath are the basal dendrites, apical dendrites are colored purple. (A) The original method of NeuroM finds a single apical point which is actually a bifurcation of the trunk. (B) Further developing the method, multiple apical points can be found. (C) The apical dendritic sections are classified. Blue: trunk, yellow: oblique dendrites, green: tuft sections.

and standard deviation of the attenuation features at the three different input distances. The digitized data and the script that calculates the feature means and standard deviations, and creates the JSON file are available here: https://github.com/sasaray/HippoUnit_demo/tree/master/target_features/Examples_on_creating_JSON_files/Magee2000-PSP_att/.

In this test the apical trunk receives excitatory post-synaptic current (EPSC)-shaped current stimuli at locations of different distances from the soma. The maximum depolarization caused by the input is extracted at the soma and divided by the maximum depolarization at the location of the stimulus to get the soma/dendrite attenuation values that are then averaged in

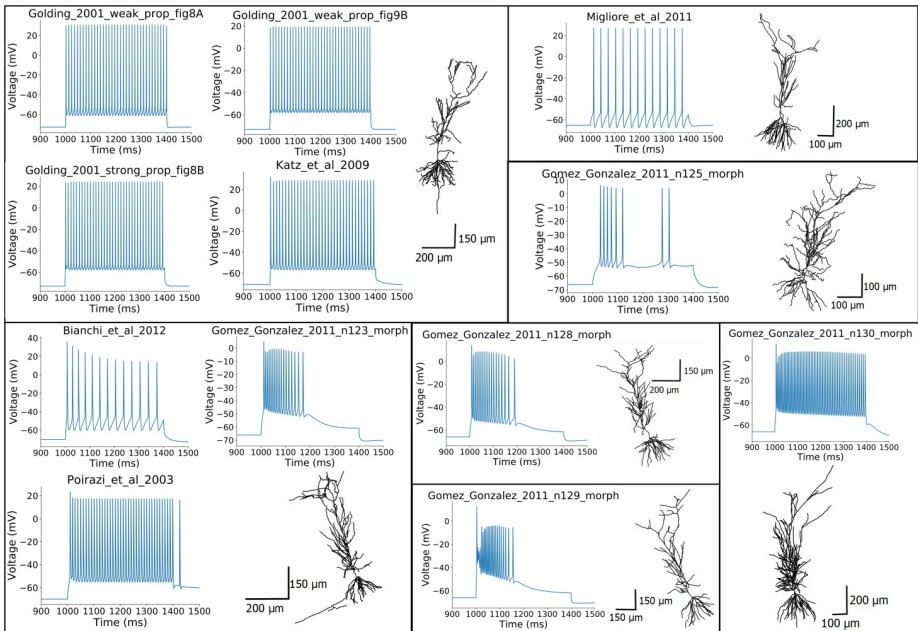

**Fig 2. The morphologies of the different models tested and their voltage responses to a 400 ms somatic step current injection of 0.6 nA amplitude.** (Some of the models share the same morphology, while the Gómez González et al. 2011 model was adjusted to five different morphologies).

distance ranges of 100, 200, 300 ± 50 μm and compared to the experimental data. The distances and tolerance are defined in the configuration file and must agree with how the *observation* data were generated.

The test uses a trunk section list, which needs to be specified in the NEURON HOC model (or the test generates one if the `find_section_lists` variable of the `ModelLoader` is set to True–see the section 'Classify apical sections of pyramidal cells' below) to find the dendritic locations to be stimulated. Randomly selected dendritic locations are used because the distance ranges that are evaluated cover almost the whole length of the trunk of a pyramidal cell. The probability of selecting a given dendritic segment is set to be proportional to its length. The number of dendritic segments examined can be chosen by the user by setting the `num_of_dend_locations` argument of the test. The random seed (also an argument of the test) must be kept constant to make the selection reproducible. If a given segment is selected multiple times (or it is closer than 50 μm or further than 350 μm), a new random number is generated. If the number of locations to be selected is more than the number of trunk segments available in the model, all the segments are selected.

The *Exp2Syn* synaptic model of NEURON with a previously calculated weight is used to stimulate the dendrite. The desired EPSC amplitude and time constants are given in the input configuration file according to the experimental protocol. To get the proper synaptic weight, first the stimulus is run with weight = 0. The last 10% of the trace is averaged to get the resting membrane potential (Vm). Then the synaptic weight required to induce EPSCs with the experimentally determined amplitude is calculated according to Eq 1:

$$weight = -EPSC_amp/Vm \qquad (1)$$

where EPSC_amp is read from the `config` dictionary, and the synaptic reversal potential is assumed to be 0 mV.

To get the somatic and dendritic maximum depolarization from the voltage traces, the baseline trace (weight = 0) is subtracted from the trace recorded in the presence of the input. To get the attenuation ratio the maximum value of the somatic depolarization is divided by the maximum value of the dendritic depolarization.

To calculate the feature scores the soma/dendrite attenuation values are first averaged over the distance ranges to be compared to the experimental data to get the feature Z-scores. The final score is the average of the feature scores calculated at the different dendritic locations.

**The Oblique Integration Test.** This test evaluates the signal integration properties of radial oblique dendrites, determined by providing an increasing number of synchronous (0.1 ms between inputs) or asynchronous (2 ms between inputs) clustered synaptic inputs. The experimental mean and standard error (SE) of the features examined are available in the paper of Losonczy and Magee [33] and are read from a JSON file into the *observation* dictionary of the test. The SE values are then converted to standard deviation values. The following features are tested: voltage threshold for dendritic spike initiation (defined as the expected somatic depolarization at which a step-like increase in peak dV/dt occurs); proximal threshold (defined the same way as above, but including only those results in the statistics where the proximal part of the examined dendrite was stimulated); distal threshold; degree of nonlinearity at threshold; suprathreshold degree of nonlinearity; peak derivative of somatic voltage at threshold; peak amplitude of somatic EPSP; time to peak of somatic EPSP; degree of nonlinearity in the case of asynchronous inputs.

The test automatically selects a list of oblique dendrites that meet the criteria of the experimental protocol, based on a section list containing the oblique dendritic sections (this can either be provided by the HOC model, or generated automatically if the

`find_section_lists` variable of the `ModelLoader` is set to True–see the section 'Classify apical sections of pyramidal cells' below). For each selected oblique dendrite a proximal and a distal location is examined. The criteria for the selection of dendrites, which were also applied in the experiments, are the following. The selected oblique dendrites should be terminal dendrites (they have no child sections) and they should be at most 120 μm from the soma. This latter criterion can be changed by the user by changing the value of the `ModelLoader`'s *max_dist_from_soma* variable, and it can also increase automatically if needed. In particular, if no appropriate oblique is found up to the upper bound provided, the distance is increased iteratively by 15 μm, but not further than 190 μm.

Then an increasing number of synaptic inputs are activated at the selected dendritic locations separately, while recording the local and somatic voltage response. HippoUnit provides a default synapse model to be used in the `ObliqueIntegrationTest`. If the *AMPA_name*, and *NMDA_name* variables are not set by the user, the default synapse is used. In this case the AMPA component of the synapse is given by the built-in Exp2Syn synapse of NEURON, while the NMDA component is defined in an NMODL (.mod) file which is part of the HippoUnit package. This NMDA receptor model uses a Jahr-Stevens voltage dependence [47] and rise and decay time constants of 3.3 and 102.38 ms, respectively. The time constant values used here are temperature- (Q10-) corrected values from [42]. Q10 values for the rise and decay time constants were 2.2 [48] and 1.7 [49], respectively. The model's own AMPA and NMDA receptor models can also be used in this test if their NMODL files are available and compiled among the other mechanisms of the model. In this case the `AMPA_name`, and `NMDA_name` variables need to be provided by the user. The time constants of the built-in Exp2Syn AMPA component and the AMPA/NMDA ratio can be adjusted by the user by setting the *AMPA_tau1*, *AMPA_tau2* and `AMPA_NMDA_ratio` parameter of the `ModelLoader`. The default AMPA/NMDA ratio is 2.0 from [42], and the default AMPA_tau1 and AMPA_tau2 are 0.1 ms and 2.0 ms, respectively [29,30].

To test the Poirazi et al. 2003 model using its own receptor models, we also had to implement a modified version of the synapse functions of the `ModelLoader` that can deal with the different (pointer-based) implementation of synaptic activation in this model. For this purpose, a child class was implemented that inherits from the `ModelLoader`. This modified version is not part of the official HippoUnit version, because this older, more complicated implementation of synaptic models is not generally used anymore; however, this is a good example on how one can modify the capability methods of HippoUnit to match their own models or purposes. The code for this modified `ModelLoader` is available here: https://github.com/KaliLab/HippoUnit_demo/blob/master/ModelLoader_Poirazi_2003_CA1.py.

The synaptic weights for each selected dendritic location are automatically adjusted by the test using a binary search algorithm so that the threshold for dendritic spike generation is 5 synchronous inputs–which was the average number of inputs that had to be activated by glutamate uncaging to evoke a dendritic spike in the experiments [33]. This search runs in parallel for all selected dendritic locations. The search interval of the binary search and the initial step size of the searching range can be adjusted by the user through the `c_minmax` and `c_step_start` variables of the `ModelLoader`. During the iterations of the algorithm the step size may decrease if needed; a lower threshold for the step size (`c_step_stop` variable of the `ModelLoader`) must be set to avoid infinite looping. Those dendritic locations where this first dendritic spike generates a somatic action potential, or where no dendritic spike can be evoked, are excluded from further analysis. To let the user know, this information is displayed on the output and also printed into the log file saved by the test. Most of the features above are extracted at the threshold input level (5 inputs).

The final score of this test is the average of the feature scores achieved by the model for the different features; however, a T-test analysis is also available as a separate score type for this test.

## Parallel computing

Most of the tests of HippoUnit require multiple simulations of the same model, either using stimuli of different intensities or at different locations in the cell. To run these simulations in parallel and save time, the Python `multiprocessing.Pool` module is used. The size of the pool can be set by the user. Moreover, all NEURON simulations are performed in multiprocessing pools to ensure that they run independently of each other, and to make it easy to erase the models after the process has finished. This is especially important in the case of HOC templates in order to avoid previously loaded templates running in the background and the occurrence of 'Template cannot be redefined' errors when the same model template is loaded again.

## Classifying the apical sections of pyramidal cells

Some of the validation tests of HippoUnit require lists of sections belonging to the different dendritic types of the apical tree (main apical trunk, apical tuft dendrites, and radial oblique dendrites). To classify the dendrites NeuroM [50] is used as a base package. NeuroM contains a script that, starting from the tuft (uppermost dendritic branches in Fig 1) endpoints, iterates down the tree to find a single common ancestor. This is considered as the apical point. The apical point is the upper end of the main apical dendrite (trunk), from where the tuft region arises. Every dendrite branching from the trunk below this point is considered an oblique dendrite.

However, there are many CA1 pyramidal cell morphologies where the trunk bifurcates close to the soma to form two or even more branches. In these cases the method described above finds this proximal bifurcation point as the apical point (see Fig 1A). To overcome this issue, we worked out and implemented a method to find multiple apical points by iterating the function provided by NeuroM. In particular, if the initial apical point is closer to the soma than a pre-defined threshold, the function is run again on subtrees of the apical tree where the root node of the subtree is the previously found apical point, to find apical points on those subtrees (see Fig 1B). When (possibly after multiple iterations) apical points that are far enough from the soma are found, NeuroM is used to iterate down from them on the parent sections, which will be the trunk sections (blue dots in Fig 1C). Iterating up, the tuft sections are found (green dots in Fig 1C), and the other descendants of the trunk sections are considered to be oblique dendrites (yellow dots in Fig 1C). Once all the sections are classified, their NeuroM coordinates are converted to NEURON section information for further use.

We note that this function can only be used for hoc models that load their morphologies from a separate morphology file (e.g., ASC, SWC) as NeuroM can only deal with morphologies provided in these standard formats. For models with NEURON morphologies implemented directly in the hoc language, the SectionLists required by a given test should be implemented within the model.

## Models from literature

In this paper we demonstrate the utility of the HippoUnit validation test suite by applying its tests to validate and compare the behavior of several different detailed rat hippocampal CA1 pyramidal cell models available on ModelDB [18]. For this initial comparison we chose models published by several modeling groups worldwide that were originally developed for various

purposes. The models compared were the following: the Golding et al., 2001 model [27] (ModelDB accession number: 64167), the Katz et al., 2009 model [51] (ModelDB accession number: 127351), the Migliore et al., 2011 model [52] (ModelDB accession number: 138205), the Poirazi et al., 2003 model [6,53] (ModelDB accession number: 20212), the Bianchi et al., 2012 model [25] (ModelDB accession number: 143719), and the Gómez González et al., 2011 [54] model (ModelDB accession number: 144450).

Models from literature that are published on ModelDB typically implement their own simulations and plots to make it easier for users and readers to reproduce and visualize the results shown in the corresponding paper. Therefore, to be able to test the models described above using our test suite, we needed to create standalone versions of them. These standalone versions do not display any GUI, or contain any built-in simulations and run-time modifications, but otherwise their behavior should be identical to the published version of the models. We also added section lists of the radial oblique and the trunk dendritic sections to those models where this was not done yet, as some of the tests require these lists. To ensure that the standalone versions have the same properties as the original models, we checked their parameters after running their built-in simulations (in case including any run-time modifications), and made sure they match the parameters of the standalone version. The modified models used for running validation tests are available in this GitHub repository: https://github.com/KaliLab/HippoUnit_demo.

## Results

### The HippoUnit validation suite

HippoUnit (https://github.com/KaliLab/hippounit) is an open source test suite for the automatic and quantitative evaluation of the behavior of neural single cell models. The tests of HippoUnit automatically perform simulations that mimic common electrophysiological protocols on neuronal models to compare their behavior with quantitative experimental data using various feature-based error functions. Current validation tests cover somatic (subthreshold and spiking) behavior as well as signal propagation and integration in the dendrites. These tests were chosen because they collectively cover diverse functional aspects of cellular behavior that have been thoroughly investigated in experimental and modeling studies, and particularly because the necessary experimental data were available in sufficient quality and quantity. However, we note that the currently implemented tests, even in combination, probably do not fully constrain the behavior of the cell under all physiological conditions, and thus the test suite can be further improved by including additional tests and more experimental data. The tests were developed using data and models for rat hippocampal CA1 pyramidal cells. However, most of the tests are directly applicable to or can be adapted for other cell types if the necessary experimental data are available; examples of this will be presented in later sections.

HippoUnit is implemented in the Python programming language, and is based on the SciUnit [19] framework for testing scientific models. The current version of HippoUnit is capable of handling single cell models implemented in the NEURON simulator, provided that they do not apply any runtime modification, do not have a built-in graphical user interface, and do not automatically perform simulations. Meeting these conditions may require some modifications in the published code of the model. Once such a "standalone" version of the model is available, the tests of HippoUnit can be run by adapting and using the example Jupyter notebooks described in S1 Appendix, without any further coding required from the user. In principle, neural models developed using other software tools can also be tested by HippoUnit; however, this requires the re-implementation by the user of the interface functions that allow HippoUnit to run the necessary simulations and record their output (see the Methods section for more details).

In the current tests of HippoUnit, once all the necessary simulations have been performed and the responses of the model have been recorded, electrophysiological features are extracted from the voltage traces, and the discrepancy between the model's behavior and the experiment is computed by comparing the feature values with those extracted from the experimental data (see Methods). Biological variability is taken into account by measuring the difference between the feature value for the model and the mean of the feature in the experiments in units of the standard deviation for that particular feature observed in the experiments. For simplicity, we refer to the result of this comparison as the feature score; however, we note that there are many possible sources of such discrepancy including, among others, experimental artefacts and noise, shortcomings of the models, and differences between the conditions assumed by the models and those in the actual experiments (see the Discussion for more details). The final score of a given test achieved by a given model is given by the average (or, in some cases, the sum) of the feature scores for all the features evaluated by the test.

While the main output of the tests is the final score, which allows the quantitative comparison of the models' behavior to experimental data, it is important to emphasize that it should never be blindly accepted. A high final score does not necessarily mean that the model is bad–it may also indicate an issue with the data, a mismatch between experimental conditions and modeling assumptions, or some problem with the implementation of the test itself (see the Discussion for further details). For this reason, and also to provide more insight into how the scores were obtained, the tests of HippoUnit typically provide a number of other useful outputs (see Methods), including figures that visualize the model's behavior through traces and plot the feature and feature score values compared to the experimental data. It is always strongly recommended to look at the traces and other figures to get a fuller picture of the model's response to the stimuli, which helps with the correct interpretation of validation results. Such closer inspection also makes it possible to detect possible test failures, when the extraction of certain features does not work correctly for a given model.

HippoUnit can also take advantage of the parallel execution capabilities of modern computers. When tests require multiple simulations of the same model using different settings (e.g., different stimulation intensities or different stimulus locations in the cell), these simulations are run in parallel, which can make the validation process substantially faster, depending on the available computing resources.

One convenient way of running a test on a model is to use an interactive computational notebook, such as the Jupyter Notebook [55], which enables the combination of program codes to be run (we used Python code to access the functionality of HippoUnit), the resulting outputs (e.g. figures, tables, text) and commentary or explanatory text in a single document. Therefore, we demonstrate the usage of HippoUnit through this method (See S1 Appendix and https://github.com/KaliLab/HippoUnit_demo).

## Comparison of the behavior of rat hippocampal CA1 pyramidal cell models selected from the literature

We selected six different publications containing models of rat hippocampal CA1 pyramidal cells whose implementations for the NEURON simulator were available in the ModelDB database. Our aim was to compare the behavior of every model to the experimental target data using the tests of HippoUnit, which also allowed us to compare the models to each other, and to test their generalization performance in paradigms that they were not originally designed to capture. These models differ in their complexity regarding the number and types of ion channels that they contain, and they were built for different purposes.

The Golding et al., 2001 model [27] was developed to show the dichotomy of the back-propagation efficacy and the amplitudes of the back-propagating action potentials at distal trunk regions in CA1 pyramidal cells and to make predictions on the possible causes of this behavior. It contains only the most important ion channels (Na, $K_{DR}$, $K_A$) needed to reproduce the generation and propagation of action potentials. [26]

The Katz et al., 2009 model [51] is based on the Golding et al. 2001 model and was built to investigate the functional consequences of the distribution of strength and density of synapses on the apical dendrites that they observed experimentally, for the mode of dendritic integration.

The Migliore et al., 2011 model [52] was used to study schizophrenic behavior. It is based on earlier models of the same modeling group, which were used to investigate the initiation and propagation of action potentials in oblique dendrites, and have been validated against different electrophysiological data.

The Poirazi et al., 2003 model [6,53] was designed to clarify the issues about the integrative properties of thin apical dendrites that may arise from the different and sometimes conflicting interpretations of available experimental data. This is a quite complex model in the sense that it contains a large number of different types of ion channels, whose properties were adjusted to fit in vitro experimental data, and it also contains four types of synaptic receptors.

The Bianchi et al., 2012 model [25] was designed to investigate the mechanisms behind depolarization block observed experimentally in the somatic spiking behavior of rat CA1 pyramidal cells. It was developed by combining and modifying the Shah et al., 2008 [56] and the Poirazi et al. 2003 models [6,53]. The former of these was developed to show the significance of axonal M-type potassium channels.

The Gómez González et al., 2011 [54] model is based on the Poirazi et al. 2003 model and it was modified to replicate the experimental data of [33] on the nonlinear signal integration of radial oblique dendrites when the inputs arrive in a short time window.

A common property of these models is that their parameters were set using manual procedures with the aim of reproducing the behavior of real rat CA1 PCs in one or a few specific paradigms. As some of them were built by modifying and further developing previous models, these share the same morphology (see Fig 2). On the other hand, the model of Gómez González et al. 2011 was adjusted to 5 different morphologies, which were all tested. In the case of the Golding et al. 2001 model, we tested three different versions (shown in Figs 8A, 8B and 9A of the corresponding paper [27]) that differ in the distribution of the sodium and the A-type potassium channels, and therefore the back-propagation efficacy of the action potentials. The morphologies and characteristic voltage responses of all the models used in this comparison are displayed in Fig 2.

Running the tests of HippoUnit on these models we took into account the original settings of the simulations of the models, and set the `v_init` (the initial voltage when the simulation starts), and the `celsius` (the temperature at which the simulation is done) variables accordingly. For the Bianchi et al 2012 model we used variable time step integration during all the simulations, as it was done in the original modeling study. For the other models a fixed time step were used (dt = 0.025 ms).

**Somatic Features Test.**   Using the Somatic Features Test of HippoUnit, we compared the behavior of the models to features extracted from the patch clamp dataset, as each of the tested models was apparently constructed using experimental data obtained from patch clamp recordings as a reference. After performing a review of the relevant literature, we concluded that the patch clamp dataset is in good agreement with experimental observations available in the literature (see Table 1 in Methods), and will be used as a representative example in this study.

In the patch clamp recordings, both the depolarizing and the hyperpolarizing current injections were 300 ms long and 0.05, 0.1, 0.15, 0.2, 0.25 nA in amplitude. Because during these recordings the cells were stimulated with relatively low amplitude current injections, some of the examined models (Migliore et al. 2011, Gómez González et al. 2011 n125 morphology) did not fire even for the highest amplitude tested. Some other models started to fire for higher current intensities than it was observed experimentally. In these cases the features that describe action potential shape or timing properties cannot be evaluated for the given model (for the current amplitudes affected). Therefore, besides the final score achieved by the models on this test (the average Z-score for the successfully evaluated features–see Methods for details) that shows the discrepancy of the models' behavior and the experimental observations regarding the successfully evaluated features, we also consider the proportion of the successfully evaluated features as an important measure of how closely the model matches this specific experimental dataset. This information, along with the names of the features that cannot be evaluated for the given model, are provided as outputs of the test, and should be considered when making conclusions on the model's performance. This is another example where looking at only the final score may not be enough to determine whether the model meets the requirements of the user, and shows how the other outputs of the tests can help the interpretation of the results.

Fig 3 shows how the extracted feature values of the somatic response traces of the different models fit the experimental values. It is clear that the behavior of the different models is very diverse. Each model captures some of the experimental features but shows a larger discrepancy for others.

The resting membrane potential (*voltage_base*) for all of the models was apparently adjusted to a more hyperpolarized value than in the experimental recordings we used for our comparison, and most of the models also return to a lower voltage value after the step stimuli (*steady_state_voltage*). An exception is the Poirazi et al. 2003 model, where the decay time constant after the stimulus is unusually high (this feature is not included in Fig 3, but the slow decay can be seen in the example trace in Fig 2, and detailed data are available here: https://github.com/KaliLab/HippoUnit_demo). The voltage threshold for action potential generation (*AP_begin_voltage*) is lower than the experimental value for most of the models (that were able to generate action potentials in response to the examined current intensities), but it is higher than the experimental value for most versions of the Gómez González et al. 2011 model. For negative current steps most of the models gets more hyperpolarized (*voltage_deflection*) (the most extreme is the Gómez González et al. 2011 model with the n129 morphology), while the Gómez González et al. 2011 model with the n125 morphology and the Migliore et al. 2011 model get less hyperpolarized than it was observed experimentally. The sag amplitudes are also quite high for the Gómez González et al. 2011 n129, and n130 models, while the Katz et al. 2009, and all versions of the Golding et al. 2001 models basically have no hyperpolarizing sag.

It is quite conspicuous how much the amplitude of the action potentials (*APlast_amp*, *AP_amplitude*, *AP2_amp*) differs in the Gómez González et al. 2011 models from the experimental values and from the other models as well. The Katz et al. 2009 and one of the versions ("Fig 8A") of the Golding et al. 2001 model have slightly too high action potential amplitudes, and these models have relatively small action potential width (*AP_width*). On the other hand, the rising phase (*AP_rise_time*, *AP_rise_rate*) of the Katz et al. 2009 model appears to be too slow.

Looking at the inverse interspike interval (*ISI*) values, it can be seen that the experimental spike trains show adaptation in the ISIs, meaning that the first ISI is smaller (the inverse ISI is higher) than the last ISI for the same current injection amplitude. This behavior can be observed in the case of the Katz et al. 2009 model, three versions (n128, n129, n130

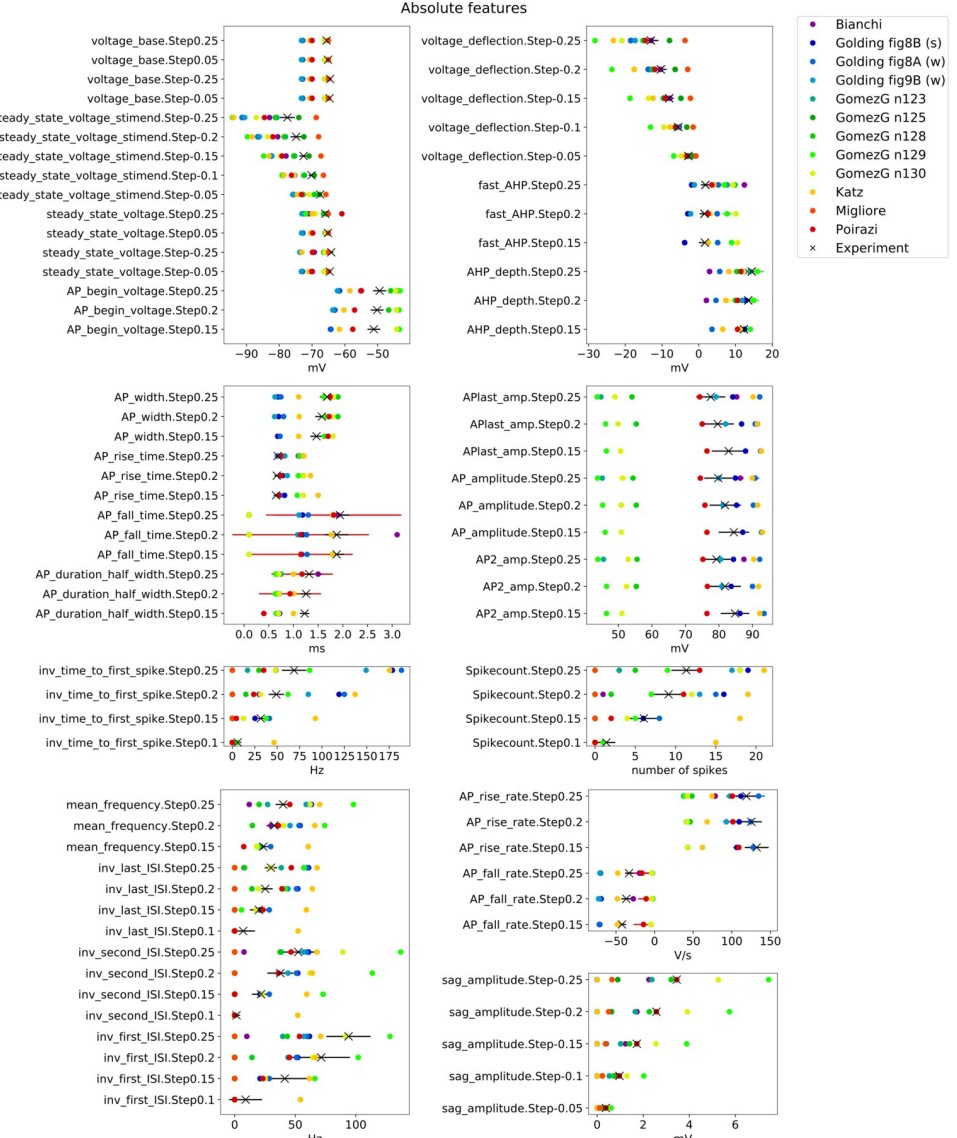

**Fig 3. Feature values from the Somatic Features Test of HippoUnit applied to several published models.** Absolute feature values extracted (using the electrophys Feature Extraction Library (eFEL)) from the voltage responses of the models to somatic current injections of varying amplitude, compared to mean experimental values (black X) that were extracted from the patch clamp dataset. Black solid, horizontal lines indicate the experimental standard deviation. Colored solid, horizontal lines typically show the standard deviation of spiking features of models, where the feature value of each action potential in the voltage trace is extracted and averaged. Feature names (y axis labels) are indicated as they are used in eFEL combined with the step current injection amplitude. Not all the evaluated features are shown here. (The (s) and (w) notations of the Golding et al. 2001 models in the legend indicate the strong and weak propagating versions of the model).

morphology) of the Gómez González et al. 2011 model, but cannot really be seen in the Bianchi et al. 2011, the Poirazi et al. 2003 and the three versions of the Golding et al. 2001 models. At first look it may seem contradictory that in the case of the Gómez González et al. 2011 model version n129 morphology the spike counts are quite low, while the mean frequency and the inverse ISI values are high. This is because the soma of this model does not fire over the whole period of the stimulation, but starts firing at higher frequencies, then stops firing for rest of the

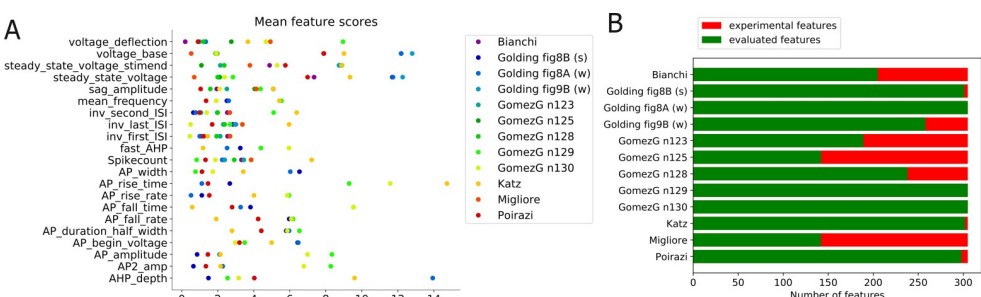

**Fig 4. Evaluation of results from the Somatic Features Test of HippoUnit applied to published models.** (A) Mean feature scores (the difference between the model's feature value and the experimental mean in units of the experimental SD) of the different models. Feature score values are averaged over the different input step amplitudes. (B) The bars represent the number of features that were attempted to be evaluated for the models (i.e., the number of features extracted from the experimental patch clamp dataset). The number of successfully evaluated features for the various models is shown in green, and the number of features that could not be evaluated for a particular model is shown in red. Features that are not evaluated successfully are most often spiking features at step amplitudes for which the tested model does not fire action potentials.

stimulus (see Fig 2). The Katz et al. 2009 model fires quite a high number of action potentials (*Spikecount*) compared to the experimental data, at a high frequency.

In the experimental recordings there is a delay before the first action potential is generated, which becomes shorter with increasing current intensity (indicated by the *inv_time_to_first_-spike* feature that becomes larger with increasing input intensity). In most of the models this behavior can be observed, albeit to different degrees. The Katz et al. 2009 model has the shortest delays (highest *inv_time_to_first_spike* values), but the effect is still visible.

To quantify the difference between the experimental dataset and the simulated output of the models, these were compared using the feature-based error function (Z-Score) described above to calculate the feature score. Fig 4A shows the mean scores of the model features whose absolute values are illustrated in Fig 3 (averaged over the different current step amplitudes examined), while Fig 4B indicates the number of successfully evaluated features out of the number of features that were attempted to be evaluated. From Fig 4A it is even more clearly visible that each model fits some experimental features well but does not capture others. For example, it is quite noticeable in Fig 4A that most of the versions of the Gómez González et al. 2011 model (greenish dots) perform well for features describing action potential timing (upper part of the figure, e.g., *ISIs*, *mean_frequency*, *spikecount*), but get higher feature scores for features of action potential shape (lower part of the figure, e.g., *AP_rise_rate*, *AP_rise_time*, *AP_fall_rate*, *AP_fall_time*, *AP amplitudes*). Conversely, the Katz et al. 2009 model achieved better scores for AP shape features than for features describing AP timing. It is also worth noting that none of the feature scores for the model of Migliore et al. 2011 was higher than 4; however, looking at Fig 4B it can be seen that less than half of the experimental features were successfully evaluated in this model, which is because it does not fire action potentials for the current injection amplitudes examined here. As mentioned above the proportion of the successfully evaluated features is also an important measure of how well the behavior of the models fits the specific experimental observations, and should be taken into account.

**Depolarization Block Test.**   In the Depolarization Block Test three features are evaluated. Two of them examine the threshold current intensity to reach depolarization block. The *I_maxNumAP* feature is the current intensity at which the model fires the maximum number of action potentials, and the *I_below_depol_block* feature is the current intensity one step before the model enters depolarization block. Both are compared to the experimental $I_{th}$ feature because, in the experiment [25], the number of spikes increased monotonically with

increasing current intensity up to the current amplitude where the cell entered depolarization block during the stimulus, which led to a drop in the number of action potentials. By contrast, we experienced that some models started to fire fewer spikes for higher current intensities while still firing over the whole period of the current step stimulus, i.e., without entering depolarization block. Therefore, we introduced the two separate features for the threshold current. If these two feature values are not equal, a penalty is added to the score. The third evaluated feature is $V_{eq}$, the equilibrium potential during the depolarization block, which is calculated as the average of the membrane potential over the last 100 ms of a current pulse with amplitude 50 pA above $I\_maxNumAP$ (or 50 pA above $I\_below\_depol\_block$ if its value is not equal to $I\_maxNumAP$). Each model has a value for the $I\_maxNumAP$ feature, while those models that do not enter depolarization block are not supposed to have a value for the $I\_below\_depol\_block$ feature and the $Veq$ feature.

The results from applying the Depolarization Block Test to the models from ModelDB are shown in Fig 5. According to the test, four of the models entered depolarization block. However, by looking at the actual voltage traces provided by the test, it becomes apparent that only the Bianchi et al. 2011 model behaves correctly (which was developed to show this behavior). The other three models actually managed to "cheat" the test.

In the case of the Katz et al. 2009 and the Golding et al. 2001 "Fig 9B" models, the APs get smaller and smaller with increasing stimulus amplitude until they get so small that they do not reach the threshold for action potential detection; therefore, these APs are not counted by the

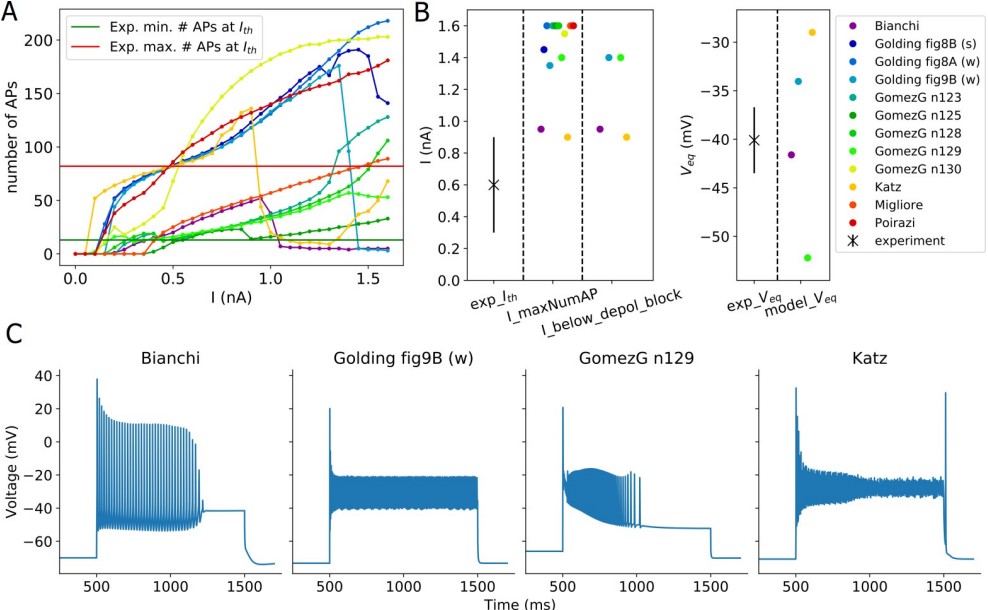

**Fig 5. Results from the Depolarization Block Test of HippoUnit applied to published models.** (A) Number of APs fired by the models in response to 1 sec long somatic current injections of increasing intensity. (B) Depolarization block feature values extracted from the voltage responses of the models compared to the experimental observations. $exp\_I_{th}$ is the mean (SD is indicated with a solid line) of the experimentally observed threshold current amplitude to reach depolarization block. In the test two separate features are compared to the experimental threshold value: The $I\_maxNumAP$ feature is the current intensity at which the model fires the maximum number of action potentials, and the $I\_below\_depol\_block$ feature is the current intensity one step before the model enters depolarization block. According to the experimental observation, these two values are supposed to be the same, but for models, they may differ, in which case a penalty is added to the final score (see the text for more details). The $V_{eq}$ is the equilibrium potential to which the somatic voltage settles after entering depolarization block. (C) Voltage traces of different models that were recognized by the test as depolarization block. Note that only the Bianchi et al. 2012 model actually entered depolarization block, the others "cheated" the test (see the main text for more details).

test and $V_{eq}$ is also calculated. The Gómez González et al. 2011 model adjusted to the n129 morphology does not fire during the whole period of the current stimulation for a wide range of current amplitudes (see Fig 2). Increasing the intensity of the current injection it fires an increasing number of spikes, but always stops after a while before the end of the stimulus. On the other hand, there is a certain current intensity after which the model starts to fire fewer action potentials, and which is thus detected as *I_maxNumAP* by the test. Because no action potentials can be detected during the last 100 ms of the somatic response one step above the detected "threshold" current intensity, the model is declared to have entered depolarization block, and a $V_{eq}$ value is also extracted.

In principle, it would be desirable to modify the test so that it correctly rejects the three models above. However, the models described above shows so similar behavior to depolarization block that is hard to distinguish using automatic methods. Furthermore, we have made substantial efforts to make the test more general and applicable to a wide variety of models with different behavior, and we are concerned that defining and adding further criteria to the test to deal with these specific cases would be an ad hoc solution, and would possibly cause further 'cheats' when applied to other models with unexpected behavior. These cases underline the importance of critically evaluating the full output (especially the figures of the recorded voltage traces) of the tests rather than blindly accepting the final scores provided.

**Back-propagating AP Test.**   This test first finds all the dendritic segments that belong to the main apical dendrite of the model and which are 50, 150, 250, 350 ± 20 μm from the soma, respectively. Then a train of action potentials of frequency around 15 Hz is triggered in the soma by injecting a step current of appropriate amplitude (as determined by the test), and the amplitudes of the first and last action potentials in the train are measured at the selected locations. In the Bianchi et al. 2012 and the Poirazi et al. 2003 models (which share the same morphology, see Fig 2) no suitable trunk locations could be found in the most proximal (50 ± 20 μm) and most distal (350 ± 20 μm) regions. This is because this morphology has quite long dendritic sections that are divided into a small number of segments. In particular, the first trunk section (apical_dendrite[0]) originates from the soma, is 102.66 μm long, and has only two segments. The center of one of them is 25.67 μm far from the soma, while the other is already 77 μm away from the soma. None of these segments belongs to the 50 ± 20 μm range, and therefore they are not selected by the test. The n123 morphology of the Gómez González et al. 2011 model has the same shape (Fig 2), but in this case the segments are different, and therefore it does not share the same problem.

At the remaining, successfully evaluated distance ranges in the apical trunk of the Bianchi et al. 2012 model, action potentials propagate very actively, barely attenuating. For the *AP1_amp* and *APlast_amp* features at these distances, this model has the highest feature score (Fig 6), while the Poirazi et al. 2003 model performs quite well.

The Golding et al. 2001 model was designed to investigate how the distribution of ion channels can affect the back-propagation efficacy in the trunk. The two versions of the Golding et al. 2001 model ("Fig 8A" and "Fig 9B" versions) which are supposed to be weakly propagating according to the corresponding paper [27], are also weakly propagating according to the test. However, the difference between their strongly and weakly propagating feature scores is not too large (Fig 6), which is probably caused by the much smaller standard deviation value of the experimental data for the weakly propagating case. Although the amplitudes of the first action potentials of these two models fit the experimental data relatively well, they start to decline slightly closer to the soma than it was observed experimentally, as the amplitudes are already very small at 250 ± 20 μm (Fig 6). (In Fig 6 the data corresponding to these two versions of the model are almost completely overlapping for more distal regions.) The amplitudes for the last action potential fit the data well, except in the most proximal regions (see the

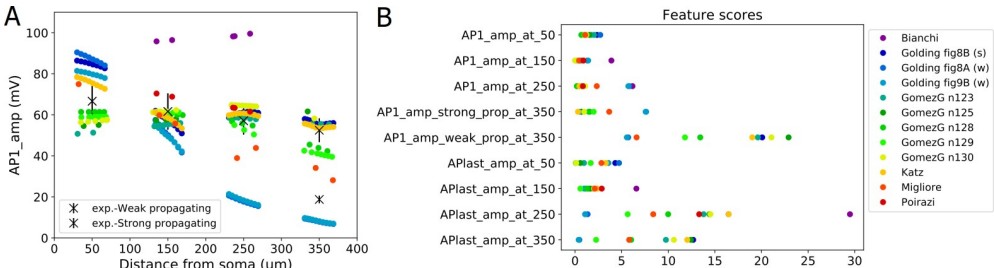

**Fig 6. Results from the Back-propagating AP Test of HippoUnit applied to published models.** (A) The amplitudes of the first back-propagating action potentials (in a train of spikes with frequency around 15 Hz evoked by somatic current injection) as a function of recording location distance from the soma. (B) Feature scores achieved by the different models on the Back-propagating AP Test. The amplitudes of the first and last back-propagating action potentials were averaged over the distance ranges of 50, 150, 250, 350 ± 20 μm and compared to the experimental features (see Methods for more details).

relatively high feature score in Fig 6B or the detailed results here: https://github.com/KaliLab/HippoUnit_demo). For all versions of the Golding et al. 2001 model, AP amplitudes are too high at the most proximal distance range. As for the strongly propagating version of the Golding et al. 2001 model ("Fig 8B" version), the amplitude of the first action potential is too high at the proximal locations, but further it fits the data well. The amplitude of the last action potential remains too high even at more distal locations. It is worth noting that, in the corresponding paper [27], they only examined a single action potential triggered by a 5 ms long input in their simulations, and did not examine or compare to their data the properties of the last action potential in a longer spike train. Finally, we note that in all versions of the Golding et al. 2001 model a spike train with frequency around 23 Hz was evoked and examined as it turned out to be difficult to set the frequency closer to 15 Hz.

The different versions of the Gómez González et al. 2011 model behave qualitatively similarly in this test, although there were smaller quantitative differences. In almost all versions the amplitudes of the first action potential in the dendrites are slightly too low at the most proximal locations but fit the experimental data better at further locations. The exceptions are the versions with the n128 and n129 morphologies, which have lower first action potential amplitudes at the furthest locations, but not low enough to be considered as weak propagating. The amplitudes for the last action potential are too high at the distal regions but fit better at the proximal ones. The only exception is the one with morphology n129, where the last action potential attenuates more at further locations and fits the data better.

In the case of the Katz et al. 2009 model, a spike train with frequency around 40 Hz was examined, as the firing frequency increases so suddenly with increasing current intensity in this model that no frequency closer to 15 Hz could be adjusted. In this model the last action potential propagates too strongly, while the dendritic amplitudes for the first action potential are close to the experimental values.

In the Migliore et al. 2011 model the amplitudes for the last action potential are too high, while the amplitude of the first back-propagating action potential is too low at locations in the 250 ± 20 μm and 350 ± 20 μm distance ranges.

Finally, all the models that we examined were found to be strongly propagating by the test, with the exception of those versions of the Golding et al. 2001 model that were explicitly developed to be weakly propagating.

**PSP Attenuation Test.**   In this test the extent of the attenuation of the amplitude of an excitatory post-synaptic potential (EPSP) is examined as it propagates towards the soma from different input locations in the apical trunk. The Katz et al. 2009, the Bianchi et al. 2012, and

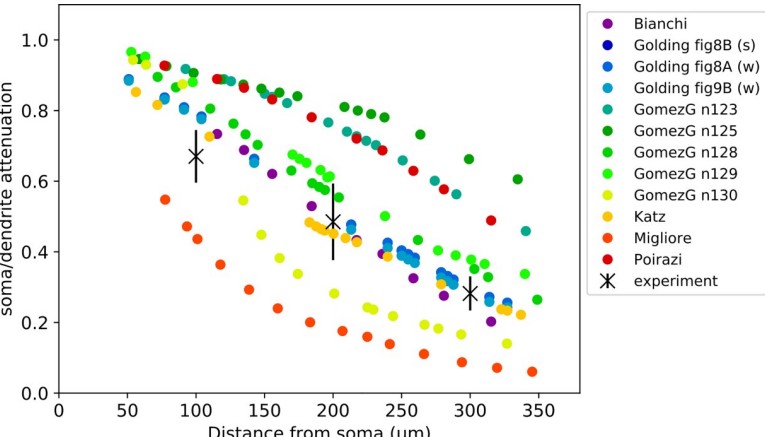

**Fig 7. Results from the PSP Attenuation Test of HippoUnit applied to published models.** Soma/dendrite EPSP attenuation as a function of the synaptic input distance from the soma in the different models.

all versions of the Golding et al. 2001 models perform quite well in this test ([Fig 7]). The various versions of the Golding et al. 2001 model are almost identical in this respect, which is not surprising as they differ only in the distribution of the sodium and A-type potassium channels. This shows that, as we would expect, these properties do not have much effect on the propagation of relatively low-amplitude signals such as unitary PSPs. Interestingly, the different versions of the Gómez González et al. 2011 model, with different morphologies, behave quite differently, which shows that this behavior can depend very much on the morphology of the dendritic tree.

**Oblique Integration Test.** This test probes the integration properties of the radial oblique dendrites of rat CA1 pyramidal cell models. The test is based on the experimental results described in [33]. In this study, the somatic voltage response was recorded while synaptic inputs in single oblique dendrites were activated in different spatio-temporal combinations

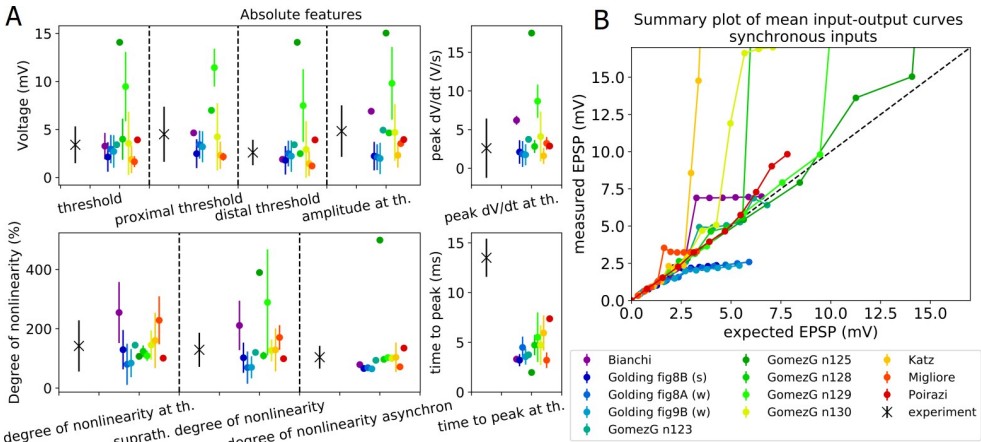

**Fig 8. Results from the Oblique Integration Test of HippoUnit applied to published models.** (A) Comparison of the responses of the models to experimental results (black X) according to features of dendritic integration. Features values are given as mean and standard deviation, as several dendritic locations of each model are tested. (B) The averaged input–output curves of all the dendritic locations examined. EPSP amplitudes are measured at the soma. Dashed line shows linearity. In models whose curve goes above the dashed line, oblique dendrites integrate synaptic inputs that are spatially and temporally clustered supralinearly.

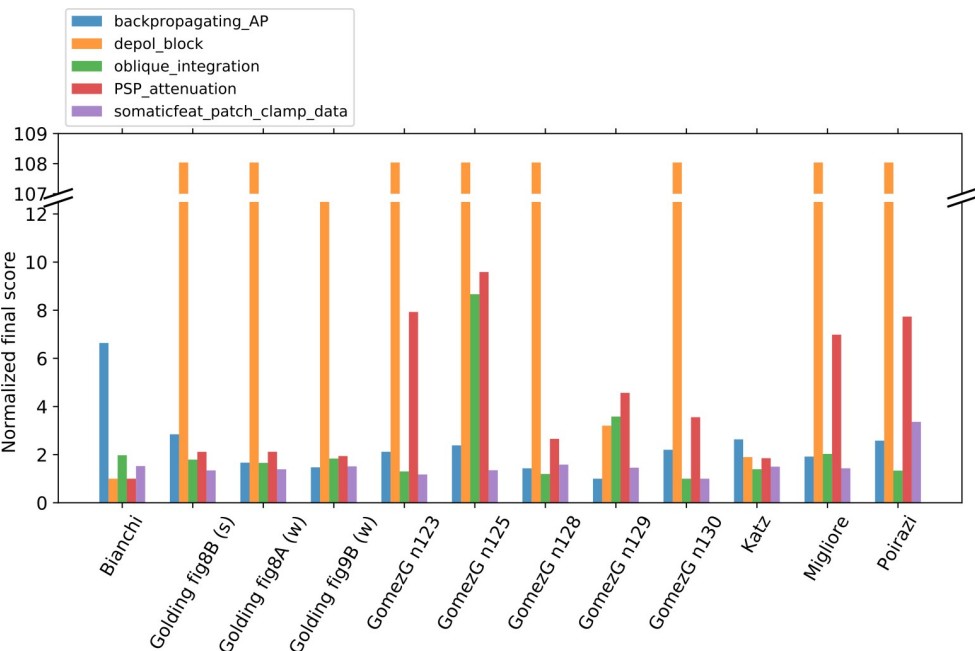

**Fig 9. Normalized final scores achieved by the different published models on the various tests of HippoUnit.** The final scores of each test are normalized by dividing the scores of each model by the best achieved score on the given test.

using glutamate uncaging. The main finding was that a sufficiently high number of synchronously activated and spatially clustered inputs produced a supralinear response consisting of a fast (Na) and a slow (NMDA) component, while asynchronously activated inputs summed linearly or sublinearly.

This test selects all the radial oblique dendrites of the model that meet the experimental criteria: they are terminal dendrites (they have no child sections) and are at most 120 μm from the soma. Then the selected dendrites are stimulated in a proximal and in a distal region (separately) using an increasing number of clustered, synchronous or asynchronous synaptic inputs to get the voltage responses of the model, and extract the features of dendritic integration. The synaptic inputs are not unitary inputs, i.e., their strength is not equivalent to the strength of one synapse in the real cell; instead, the strength is adjusted in a way that 5 synchronous inputs are needed to trigger a dendritic action potential. The intensity of the laser used for glutamate uncaging was set in a similar way in the experiments [33]. Most of the features were extracted at this just-suprathreshold level of input. We noticed that in some cases the strength of the synapse is not set correctly by the test; for example, it may happen that an actual dendritic spike does not reach the spike detection threshold in amplitude, or sometimes the EPSP may reach the threshold for spike detection without actual spike generation. The user has the ability to set the threshold used by eFEL for spike detection, but sometimes a single threshold may not work even for the different oblique dendrites (and proximal and distal locations in the same dendrites) of a single model. For consistency, we used the same spike detection threshold of -20 mV for all the models.

The synaptic stimulus contains an AMPA and an NMDA receptor-mediated component. As the default synapse, HippoUnit uses the Exp2Syn double exponential synapse built into NEURON for the AMPA component, and its own built-in NMDA receptor model, whose parameters were set according to experimental data from the literature (see the Methods

section for more details). In those models that originally do not have any synaptic component (the Bianchi et al 2011 model and all versions of the Golding et al. 2001 model) this default synapse was used. Both the Katz et al. 2009 and the Migliore et al. 2011 models used the Exp2Syn in their simulations, so in their case the time constants of this function were set to the values used in the original publications. As these models did not contain NMDA receptors, the default NMDA receptor model and the default AMPA/NMDA ratio of HippoUnit were used. The Gómez González et al 2011 and the Poirazi et al. 2003 models have their own AMPA and NMDA receptor models and their own AMPA/NMDA ratio values to be tested with.

As shown by the averaged "measured EPSP vs expected EPSP" curves in Fig 8, all three versions of the Golding et al. 2001 model have a jump in the amplitude of the somatic response at the threshold input level, which is the result of the generation of dendritic spikes. However, even these larger average responses do not reach the supralinear region, as it would be expected according to the experimental observations [33]. The reason for this discrepancy is that a dendritic spike was generated in the simulations in only a subset of the stimulated dendrites; in the rest of the dendrites tested, the amplitude of the EPSPs went above the spike detection threshold during the adjustment of the synaptic weight without actually triggering a dendritic spike, which led to the corresponding synaptic strength being incorrectly set for that particular dendrite. Averaging over the results for locations with and without dendritic spikes led to an overall sublinear integration profile.

The Migliore et al. 2011 model performs quite well on this test. In this case, seven dendrites could be tested out of the ten dendrites within the correct distance range because, in the others, the dendritic spike at the threshold input level also elicited a somatic action potential, and therefore these dendrites were excluded from further testing.

In the Katz et al. 2009 model all the selected dendritic locations could be tested, and in most of them the synaptic strength could be adjusted appropriately. For a few dendrites, some input levels higher than the threshold for dendritic spike generation also triggered somatic action potentials. This effect causes the high supralinearity in the "measured EPSP vs expected EPSP" curve in Fig 8, but has no effect on the extracted features.

In the Bianchi et al. 2012 model only one dendrite could be selected, in which very high amplitude dendritic spikes were evoked by the synaptic inputs, making the signal integration highly supralinear.

In the Poirazi et al. 2003 model also only one dendrite could be selected based on its distance from the soma; furthermore, only the distal location could be tested even in this dendrite, as at the proximal location the dendritic action potential at the threshold input level generated a somatic action potential. However, at the distal location, the synaptic strength could not be set correctly. For the synaptic strength chosen by the test, the actual threshold input level where a dendritic spike is first generated is at 4 inputs, but this dendritic AP is too small in amplitude to be detected, and the response to 5 inputs is recognized as the first dendritic spike instead. Therefore, the features that should be extracted at the threshold input level are instead extracted from the voltage response to 5 inputs. In this model this results in a reduced *supralinearity* value, as this feature is calculated one input level higher than the actual threshold. In addition, for even higher input levels dendritic bursts can be observed, which causes large *supralinearity* values in the "measured EPSP vs expected EPSP" curve in Fig 8, but this does not affect the feature values.

Models from Gómez González et al. 2011 were expected to be particularly relevant for this test, as these models were tuned to fit the same data set on which this test is based. However, we encountered an important issue when comparing our test results for these models to the results shown in the paper [54]. In particular, the paper clearly indicates which dendrites were examined, and it is stated that those are at maximum 150 μm from the soma. However, when

we measured the distance of these locations from the soma by following the path along the dendrites (as it is done by the test of HippoUnit), we often found it to be larger than 150 μm. We note that when the distance was measured in 3D coordinates rather than along the dendrites, all the dendrites used by Gómez González et al. 2011 appeared to be within 150 μm of the soma, so we assume that this definition was used in the paper. As we consider the path distance to be more meaningful than Euclidean distance in this context, and this was also the criterion used in the experimental study, we consistently use path distance in HippoUnit to find the relevant dendritic segments. Nevertheless, this difference in the selection of dendrites should be kept in mind when the results of this validation for models of Gómez González et al. 2011 are evaluated.

In two versions of the Gómez González et al. 2011 model (those that were adjusted to the n123 and n125 morphologies) only one oblique dendrite matched the experimental criteria and could therefore be selected, and these are not among those that were studied by the developers of the model. In each of these cases the dendritic spike at the proximal location at the input threshold level triggered a somatic action potential, and therefore only the distal location could be tested. In the case of the n125 morphology, the dendritic spikes that appear first for just-suprathreshold input are so small in amplitude that they do not reach the spike detection threshold (-20 mV), and are thus not detected. Therefore, the automatically adjusted synaptic weight is larger than the appropriate value would be, which results in larger somatic EPSPs than expected (see Fig 8). With this synaptic weight, the first dendritic spike and therefore the jump to the supralinear region in the "measured EPSP vs expected EPSP" curve is for 4 synaptic inputs instead of 5. This is also the case in one of the two selected dendrites of the version of this model with the n128 morphology. Similarly to the Poirazi et al. 2003 model, this results in a lower *degree of nonlinearity at threshold* feature value, than it would be if the feature were extracted at the actual threshold input level (4 inputs) instead of the one which the test attempted to adjust (5 inputs). The *suprathreshold nonlinearity* feature has a high value because at that input level (6 inputs), somatic action potentials are triggered.

In the version of the Gómez González et al. 2011 model that uses the n129 morphology, 10 oblique dendrites could be selected for testing (none of them is among those that its developers used) but only 4 could be tested because, for the rest, the dendritic spike at the threshold input level already elicits a somatic action potential. The synaptic weights required to set the threshold input level to 5 are not found correctly in most cases; the actual threshold input level is at 4 or 3. Suprathreshold nonlinearity is high, because at that input level (6 inputs) somatic action potentials are triggered for some of the examined dendritic locations.

The version of the Gómez González et al. 2011 model that uses the n130 morphology achieves the best (lowest) final score on this test. In this model many oblique dendrites could be selected and tested, including two (179, 189) that the developers used in their simulations [54]. In most cases the synaptic weights are nicely found to set the threshold input level to 5 synapses. For some dendrites there are somatic action potentials at higher input levels, but that does not affect the features.

The value of the *time to peak* feature for each model is much smaller than the experimental value (Fig 8). This is because in each of the models the maximum amplitude of the somatic EPSP is determined by the fast component, caused by the appearance of the dendritic sodium spikes, while in the experimental observation this is rather shaped by the slow NMDA component following the sodium spike.

**Overall characterization and model comparison based on all tests of HippoUnit.**    In summary, using HippoUnit, we compared the behavior of several rat hippocampal CA1 pyramidal cell models available on ModelDB in several distinct domains, and found that all of these models match experimental results well in some domains (typically those that they were originally built to capture) but fit the experimental observations less precisely in others. Fig 9

summarizes the final scores achieved by the different models on the various tests (lower scores indicate a better match in all cases).

Perhaps a bit surprisingly, the different versions of the Golding et al. 2001 model showed a good match to the experimental data in all of the tests (except for the Depolarization Block Test), even though these are the simplest ones among the models in the sense that they contain the smallest number of different types of ion channels. On the other hand, these models do not perform outstandingly well on the Back-propagating Action Potential Test, although they were developed to study the mechanisms behind (the dichotomy of) action potential back-propagation, which is evaluated by this test based on the data that were published together with these models [27]. The most probable reason for this surprising observation is that, in the original study [27], only a few features of the model's response were compared with the experimental results. HippoUnit tested the behavior of the model based on a larger set of experimental features from the original study, and was therefore able to uncover differences between the model's response and the experimental data on features for which the model was not evaluated in the source publication.

The Bianchi et al. 2012 model is the only one that can produce real depolarization block within the range of input strengths examined by the corresponding test. The success of this model in this test is not surprising because this is the only model that was tuned to reproduce this behavior; on the other hand, the failure of the other models in this respect clearly shows that proper depolarization block requires some combination of mechanisms that are at least partially distinct from those that allow good performance in the other tests. The Bianchi et al. 2012 model achieves a relatively high final score only on the Back-propagating Action Potential Test, as action potentials seem to propagate too actively in its dendrites, leading to high AP amplitudes even in more distal compartments.

The Gómez González et al. 2011 models were developed to capture the same experimental observations on dendritic integration that are tested by the Oblique Integration Test of HippoUnit, but, somewhat surprisingly, some of its versions achieved quite high feature scores on this test, while others perform quite well. This is partly caused by the fact that HippoUnit often selects different dendritic sections for testing from those that were studied by the developers of these models (see above for details). The output of HippoUnit shows that the different oblique dendrites of these models can show quite diverse behavior, and beyond those studied in the corresponding paper [54], other oblique dendrites do not necessarily match the experimental observations. Some of its versions also perform relatively poorly on the PSP-Attenuation Test, similar to the Migliore et al. 2011 and the Poirazi et al. 2003 models. The Katz et al. 2009 model is not outstandingly good in any of the tests, but still achieves relatively good final scores everywhere (although its apparent good performance on the Depolarization Block Test is misleading—see detailed explanation above).

The model files that were used to test the models described above, the detailed validation results (all the output files of HippoUnit), and the Jupyter Notebooks that show how to run the tests of HippoUnit on these models are available in the following Github repository: https://github.com/KaliLab/HippoUnit_demo.

## Application of HippoUnit to models built using automated parameter optimization within the human brain project

Besides enabling a detailed comparison of published models, HippoUnit can also be used to monitor the performance of new models at various stages of model development. Here, we illustrate this by showing how we have used HippoUnit within the HBP to systematically validate detailed multi-compartmental models of hippocampal neurons developed using multi-objective parameter optimization methods implemented by the open source Blue Brain Python

Optimization Library (BluePyOpt [16]). To this end, we extended HippoUnit to allow it to handle the output of optimization performed by BluePyOpt (see Methods).

Models of rat CA1 pyramidal cells were optimized using target feature data extracted from sharp electrode recordings [3]. Then, using the Somatic Features Test of HippoUnit, we compared the behavior of the models to features extracted from this sharp electrode dataset. However, while only a subset of the features extracted by eFEL was used in the optimization (mostly those that describe the rate and timing of the spikes; e.g., the different inter-spike interval (ISI), time to last/first spike, mean frequency features), we considered all the eFEL features that could be successfully extracted from the data during validation.

In addition, sharp electrode measurements were also available for several types of interneurons in the rat hippocampal CA1 region, and models of these interneurons were also constructed using similar automated methods [3]. Using the appropriate observation file and the stimulus file belonging to it, the Somatic Features Test of HippoUnit can also be applied to these models to evaluate their somatic spiking features. The other tests of HippoUnit are currently not applicable to interneurons, mostly due to the lack of appropriate target data.

We applied the tests of HippoUnit to the version of the models published in [3], and to a later version (v4) described in Ecker et al. (2020)[57], which was intended to further improve the dendritic behavior of the models, as this is critical for their proper functioning in the network. The two sets of models were created using the same morphology files and similar optimization methods and protocols. These new optimizations differed mainly in the allowed range for the density of the sodium channels in the dendrites. For the pyramidal cell models a new feature was also introduced in the parameter optimization that constrains the amplitudes of back-propagating action potentials in the main apical dendrite. The new interneuron models also had an exponentially decreasing (rather than constant) density of Na channels, and A-type K channels with more hyperpolarized activation in their dendrites. For more details on the models, see the original publications ([3,57]).

After running all the tests of HippoUnit on both sets of models generated by BluePyOpt, we performed a comparison of the old [3] and the new versions of the models by doing a statistical analysis of the final scores achieved by the models of the same cell type on the different tests. In Fig 10 the median, the interquartile range and the full range of the final scores achieved by the two versions of the model set are compared. According to the results of the Wilcoxon signed-rank test the new version of the models achieved significantly better scores on the Back-propagating Action Potential test (p = 0.0046), on the Oblique Integration Test (p = 0.0033), and on the PSP Attenuation Test (p = 0.0107), which is the result of reduced dendritic excitability. Moreover, in most of the other cases the behavior of the models improved slightly (but not significantly) with the new version. Only in the case of the Somatic Features test applied to bAC interneurons did the new models perform slightly worse (but still quite well), and this difference was not significant (p = 0.75).

These results show the importance of model validation performed against experimental findings, especially those not considered when building the model, in every iteration during the process of model development. This approach can greatly facilitate the construction of models that perform well in a variety of contexts, help avoid model regression, and guide the model building process towards a more robust and general implementation.

## Integration of HippoUnit into the Validation Framework and the Brain Simulation Platform of the human brain project

The HBP is developing scientific infrastructure to facilitate advances in neuroscience, medicine, and computing [58]. One component of this research infrastructure is the Brain

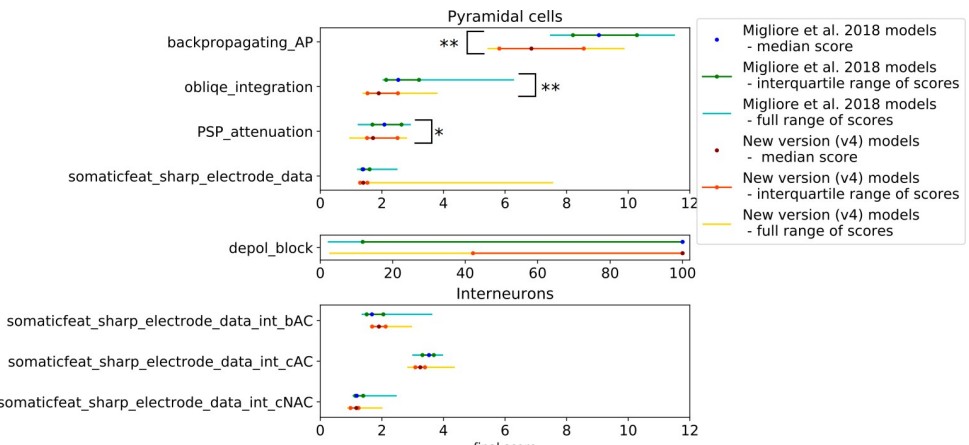

**Fig 10. Employing the tests of HippoUnit to monitor the behavior of a set of detailed data-driven models of hippocampal neurons at different stages of model development.** Models of four different cell types (pyramidal cells and continuous accommodating (int cAC), bursting accommodating (int bAC) and continuous non-accommodating (int cNAC) interneurons) of the rat hippocampal CA1 region were developed within the Human Brain Project by automated optimization using BluePyOpt. The tests of HippoUnit were used to evaluate and compare the behavior of the older (Migliore et al 2018) version and the new (v4) version of these models. The median, the interquartile range and the full range of the final scores achieved by the models of each cell type were calculated and the results of the two versions of the model set are compared. Asterisks indicate significant differences (*: p<0.05; **: p<0.01).

Simulation Platform (BSP) (https://bsp.humanbrainproject.eu), an online collaborative platform that supports the construction and simulation of neural models at various scales. As we argued above, systematic, automated validation of models is a critical prerequisite of collaborative model development. Accordingly, the BSP includes a software framework for quantitative model validation and testing that explicitly supports applying a given validation test to different models and storing the results [59]. The framework consists of a web service, and a set of test suites, which are Python modules based on the SciUnit package. As we discussed earlier, SciUnit uses the concept of capabilities, which are standardized interfaces between the models to be tested and the validation tests. By defining the capabilities to which models must adhere, individual validation tests can be implemented independently of any specific model and used to validate any compatible model despite differences in their internal structures, the language and/or the simulator used. Each test must include a specification of the required model capabilities, the location of the reference (experimental) dataset, and data analysis code to transform the recorded variables (e.g., membrane potential) into feature values that allow the simulation results to be directly and quantitatively compared to the experimental data through statistical analysis. The web services framework [59] supports the management of models, tests, and validation results. It is accessible via web apps within the HBP Collaboratory, and also through a Python client. The framework makes it possible to permanently record, examine and reproduce validation results, and enables tracking the evolution of models over time, as well as comparison against other models in the domain.

Every test of HippoUnit described in this paper has been individually registered in the Validation Framework. The JSON files containing the target experimental data for each test are stored (besides the HippoUnit_demo GitHub repository) in storage containers at the Swiss National Supercomputing Centre (CSCS), where they are publicly available. The location of the corresponding data file is associated with each registered test, so that the data are loaded automatically when the test is run on a model via the Validation Framework. As the Somatic Features Test of HippoUnit was used to compare models against five different data sets (data

from sharp electrode measurements in pyramidal cells and interneurons belonging to three different electronic types, and data obtained from patch clamp recordings in pyramidal cells), these are considered to be and have been registered as five separate tests in the Validation Framework.

All the models that were tested and compared in this study (including the CA1 pyramidal cell models from the literature and the BluePyOpt optimized CA1 pyramidal cells and interneurons of the HBP) have been registered and are available in the Model Catalog of the Validation Framework with their locations in the CSCS storage linked to them. In addition to the modifications that were needed to make the models compatible with testing with HippoUnit (described in the section "Methods–Models from literature"), the versions of the models uploaded to the CSCS container also contain an `__init__.py` file. This file implements a python class that inherits all the functions of the `ModelLoader` class of HippoUnit without modification. Its role is to make the validation of these models via the Framework more straightforward by defining and setting the parameters of the `ModelLoader` class (such as the path to the HOC and NMODL files, the name of the section lists, etc.) that otherwise need to be set after instantiating the ModelLoader (see the HippoUnit_demo GitHub repository: https://github.com/KaliLab/HippoUnit_demo/tree/master/jupyter_notebooks).

The validation results discussed in this paper have also been registered in the Validation Framework, with all their related files (output figures and JSON files) linked to them. These can be accessed using the Model Validation app of the framework.

The Brain Simulation Platform of the HBP contains several online 'Use Cases', which are available on the platform and help the users to try and use the various established pipelines. The Use Case called 'Hippocampus Single Cell Model Validation' can be used to apply the tests of HippoUnit to models that were built using automated parameter optimization within the HBP.

The Brain Simulation Platform also hosts interactive "Live Paper" documents that refer to published papers related to the models or software tools on the Platform. Live Papers provide links that make it possible to visualize or download results and data discussed in the respective paper, and even to run the associated simulations on the Platform. We have created a Live Paper (https://humanbrainproject.github.io/hbp-bsp-live-papers/2020/saray_et_al_2020/saray_et_al_2020.html) showing the results of the study presented in this paper in more detail. This interactive document provides links to all the output figures and data files resulting from the validation of the models from literature discussed here. This provides a more detailed insight into their behavior individually. Moreover, as part of this Live Paper a HippoUnit Use Case is also available in the form of a Jupyter Notebook, which guides the user through running the validation tests of HippoUnit on the models from literature that are already registered in the Framework, and makes it possible to reproduce the results presented here.

## Discussion

### Applications of the HippoUnit test suite

In this article, we have described the design, usage, and some initial applications of HippoUnit, a software tool that enables the automated comparison of the physiological properties of models of hippocampal neurons with the corresponding experimental results. HippoUnit, together with its possible extensions and other similar tools, allows the rapid, systematic evaluation and comparison of neuronal models in multiple domains. By providing the software tools and examples for effective model validation, we hope to encourage the modeling community to use more systematic testing during model development, with the aim of making the process of model building more efficient, reproducible and transparent.

One important use case for the application of HippoUnit is the evaluation and comparison of existing models. We demonstrated this by using HippoUnit to test and compare the behavior of several models of rat CA1 pyramidal neurons available on ModelDB [18] in several distinct domains against electrophysiological data available in the literature (or shared by collaborators). Besides providing independent and standardized verification of the behavior of the models, the results also allow researchers to judge which existing models show a good match to the experimental data in the domains that they care about, and thus to decide whether they could re-use one of the existing models in their own research.

Besides enabling the comparison of different models regarding how well they match a particular dataset, the tests of HippoUnit also allow one to determine the match between a particular model and several datasets of the same type. As experimental results can be heavily influenced by recording conditions and protocols, and also depend on factors such as the strain, age, and sex of the animal, it is important to find out whether the same model can simultaneously capture the outcome of different experiments, and if not, how closely it is able to match the different datasets. As an example, it would be possible using the Somatic Features Test of HippoUnit to compare the somatic behavior of a particular model to features extracted from both patch clamp and sharp electrode recordings and determine which of these is captured better by the model.

HippoUnit is also a useful tool during model development. In a typical data-driven modeling scenario, researchers decide which aspects of model behavior are relevant for them, find experimental data that constrain these behaviors, then use some of these data to build the model, and use the rest of the data to validate the model. HippoUnit and similar test suites make it possible to define quantitative criteria for declaring a model valid (ideally before modeling starts), and to apply these criteria consistently throughout model development. We demonstrated this approach through the example of detailed single cell models of rat CA1 pyramidal cells and interneurons optimized within the HBP.

Furthermore, several authors have argued for the benefits of creating "community models" [7,60,61] through the iterative refinement of models in an open collaboration of multiple research teams. Such consensus models would aim to capture a wide range of experimental observations, and may be expected to generalize (within limits) to novel modeling scenarios. A prerequisite for this type of collaborative model development is an agreement on which experimental results will be used to constrain and validate the models. Automated test suites provide the means to systematically check models with respect to all the relevant experimental data, with the aim of tracking progress and avoiding "regression," whereby previously correct model behavior is corrupted by further tuning.

Finally, the tests of HippoUnit have been integrated into the recently developed Validation Framework of the HBP, which makes it possible to collect neural models and validation tests, and supports the application of the registered tests to the registered models. Most importantly, it makes it possible to save the validation results and link them to the models in the Model Catalog, making them publicly available and traceable for the modeling community.

## Interpreting the results of HippoUnit

It is important to emphasize that a high final score on a given validation test using a particular experimental dataset does not mean that the model is not good enough or cannot be useful for a variety of purposes (including the ones it was originally developed for). The discrepancy between the target data and the model's behavior, as quantified by the validation tests, may be due to several different reasons. First, all experimental data contain noise and may have systematic biases associated with the experimental methods employed. Sometimes the

experimental protocol is not described in sufficient detail to allow its faithful reproduction in the simulations. It may also occur that a model is based on experimental data that were obtained under conditions that are substantially different from the conditions for the measurement of the validation target dataset. Using different recording techniques, such as sharp electrode or patch clamp recordings or the different circumstances of the experiments (e.g., the strain, age, and sex of the animal, or the temperature during measurement) can heavily affect the experimental results. Furthermore, the post-processing of the recorded electrophysiological data can also alter the results. For these reasons, probably no single model should be expected to achieve an arbitrarily low score on all of the validation tests developed for a particular cell type. Keeping this in mind, it is important that the modelers decide which properties of the cell type are relevant for them, and what experimental conditions they aim to mimic. Validation results should be interpreted or taken into account accordingly, and the tests themselves may need to be adapted. The interpretation of the results is aided by several additional outputs of the tests besides the final score. The traces, the extracted feature values as well as the feature scores are saved into output files and also plotted for visualization. This information is intended to help determine the strengths and weaknesses of the model and evaluate its usefulness according to the needs of the user.

The issue of neuronal variability also deserves consideration in this context. The morphology, biophysical parameters, and physiological behavior of neurons is known to be non-uniform, even within a single cell type, and this variability may be important for the proper functioning and robustness of neural circuits[62–66]. Recent models of neuronal networks have also started to take into account this variability [11,67,68]. The tests of HippoUnit account for experimental variability by measuring the distance of the feature values of the model from the experimental mean (the feature score) in units of the experimental standard deviation. This means that any feature score less than about 1 actually corresponds to behavior which may be considered "typical" in the experiments (within one standard deviation of the mean), and a feature score of 2 or 3 may still be considered acceptable for any single model. In fact, even higher values of the feature score may sometimes be consistent with the data if the experimental distribution is long-tailed rather than normal. However, such high values of the feature score certainly deserve attention as they signal a large deviation from the typical behavior observed in the experiments.

Furthermore, the acceptable feature score will generally depend on the goal of the modeling study. In particular, a study which intends to construct and examine a single model of typical experimental behavior should aim to keep all the relevant feature scores relatively low. On the other hand, when modeling entire populations of neurons, one should be prepared to accept a wider range of feature scores in some members of the model population, although the majority of the cells (corresponding to typical members of the experimental population) should still display relatively low scores. In fact, when modeling populations of neurons, one would ideally aim to match the actual distribution of neuronal features (including the mean, standard deviation, and possibly higher moments as well), and the distribution of feature scores (and actual feature values) from the relevant tests of HippoUnit actually provides the information that is necessary to compare the variability of the experimental and model cell populations.

## Uniform model formats reduce the costs of validation

Although HippoUnit is built in a way that its tests are, in principle, model-agnostic, so that the implementation of the tests does not depend on model implementation, it still required a considerable effort to create the standalone versions of the models from literature to be tested, even though all of the selected models were developed for the NEURON simulator. This is

because each model has a different file structure and internal logic that needs to be understood in order to create an equivalent standalone version. When the section lists of the main dendritic types do not exist, the user needs to create them by extensively analyzing the morphology and even doing some coding. In order to reduce the costs of systematic validation, models would need to be expressed in a format that is uniform and easy to test. As HippoUnit already has its capability functions implemented in a way that it is able to handle models developed in NEURON, the only requirement for such models is that they should contain a HOC file that describes the morphology (including the section lists for the main dendritic types of the dendritic tree) and all the biophysical parameters of the model, without any additional simulations, GUIs or run-time modifications. Currently, such a standalone version of the models is not made available routinely in publications or on-line databases, but could be added by the creators of the models with relatively little effort.

On the other hand, applying the tests of HippoUnit to models built in other languages requires the re-implementation of the capability functions that are responsible for running the simulations on the model (see Methods). In order to save the user from this effort, it would be useful to publish neuronal models in a standard and uniform format that is simulator independent and allows general use in a variety of paradigms. This would allow an easier and more transparent process of community model development and validation, as it avoids the need of reimplementation of parts of software tools (such as validation suites), and the creation of new, (potentially) non-traced software versions. This approach is already initiated for neurons and neuronal networks by the developers of NeuroML [69], NineML [70], PyNN [71], Sonata [72], and Brian [73]. Once a large set of models becomes available in these standardized formats, it will be straightforward to extend HippoUnit (and other similar test suites) to handle these models.

## Extensibility of HippoUnit

Although we were aiming to develop a test suite that is as comprehensive as possible, and that captures the most typical and basic properties of the rat hippocampal CA1 pyramidal cell, the list of features that can be tested by HippoUnit is far from complete. Upon availability of the appropriate quantitative experimental data, new tests addressing additional properties of the CA1 pyramidal cell could be included, for example, on the signal integration of the basal or the more distal apical dendrites, or on action potential initiation and propagation in the axon. Therefore, we implemented HippoUnit in a way that makes it possible to extend it by adding new tests.

As HippoUnit is based on the SciUnit package [19] it inherits SciUnits's modular structure. This means that a test is usually composed of four main classes: the test class, the model class, the capabilities class and the score class (as described in more detail in the Methods section). Thanks to this structure it is easy to extend HippoUnit with new tests by implementing them in new test classes and adding the capabilities and scores needed. The methods of the new capabilities can be implemented in the `ModelLoader` class, which is a generalized Model class for models built in the NEURON simulator, or in a newly created Model class specific to the model to be tested.

Adding new tests to HippoUnit requires adding the corresponding target experimental data as well in the form of a JSON file. The way the JSON files are created depends on the nature and source of the experimental data. In some cases the data may be explicitly provided in the text of the papers (as for the Oblique Integration and the Depolarization Block tests), therefore their JSON files are easy to make manually. Most typically, the data have to be processed to get the desired feature mean and standard deviation values and create the JSON file.

In these cases it is worth writing a script that does this automatically. Some examples on how this was done for the current tests of HippoUnit are available here: https://github.com/sasaray/HippoUnit_demo/tree/master/target_features/Examples_on_creating_JSON_files/.

As HippoUnit is open-source and is shared on GitHub, it is possible for other developers, modelers or scientists to modify or extend the test suite working on their own forks of the repository. If they would like to directly contribute to HippoUnit, a 'pull request' can be created to the main repository.

## Generalization possibilities of the tests of HippoUnit

In the current version of HippoUnit most of the validation tests can only be used to test models of rat hippocampal CA1 pyramidal cells, as the observation data come from electrophysiological measurements of this cell type and the tests were designed to follow the experimental protocols of the papers from which these data derive. However, with small modifications most of the tests can be used for other cell types, or with slightly different stimulation protocols, if there are experimental data available for the features or properties tested.

The Somatic Features Test can be used for any cell type and with any current step injection protocol even in its current form using the appropriate data and configuration files. These two files must be in agreement with each other; in particular, the configuration file should contain the parameters of the step current protocols (delay, duration, amplitude) used in the experiments from which the feature values in the data file derive. In this study this test was used with two different experimental protocols (sharp electrode measurements and patch clamp recordings that used different current step amplitudes and durations), and for testing four different cell types (rat hippocampal CA1 PC and interneurons).

In the current version of the Depolarization Block Test the properties of the stimulus (delay, duration, and amplitudes) are hard-coded to reproduce the experimental protocol used in a study of CA1 PCs [25]. However, the test could be easily modified to read these parameters from a configuration file like in the case of other tests, and then the test could be applied to other cell types if data from similar experimental measurements are available.

As the Back-propagating AP Test examines the back-propagation efficacy of action potentials in the main apical dendrite (trunk), it is mainly suitable for testing pyramidal cell models; however, it can be used for PC models from other hippocampal or cortical regions, potentially using different distance ranges of the recording sites. If different distances are used, the feature names ('AP1_amp_X' and 'APlast_amp_X', where X is the recording distance) in the observation data file and the recording distances given in the stimuli file must be in agreement. Furthermore, it would also be possible to set a section list of other dendritic types instead of the trunk to be examined by the test. This way, models of other cell types (with dendritic trees qualitatively different from those of PCs) could also be tested. The frequency range of the spike train (10–20 Hz, preferring values closest to 15 Hz) is currently hard-coded in the function that automatically finds the appropriate current amplitude, but the implementation could be made more flexible in this case as well.

The PSP Attenuation Test is quite general. Both the distances and tolerance values that determine the stimulation locations on the dendrites and the properties of the synaptic stimuli are given using the configuration file. Here again the feature names in the observation data file ('attenuation_soma/dend_x_um', where x is the distance from the soma) must fit the distances of the stimulation locations in the configuration file when one uses the tests with data from a different cell type or experimental protocol. Similarly to the Back-propagating AP Test the PSP Attenuation Test also examines the main apical dendrite (trunk), but could be altered to use section lists of other dendritic types.

The Oblique Integration Test is very specific to the experimental protocol of [33]. There is no configuration file used here, but the synaptic parameters (of the `ModelLoader` class) and the number of synapses to which the model should first generate a dendritic spike ('threshold_index' parameter of the test class) can be adjusted by the user after instantiating the `ModelLoader` and the test classes respectively. The time intervals between the inputs (synchronous (0.1 ms), asynchronous (2.0 ms)) are currently hard-coded in the test.

HippoUnit has been used mainly to test models of rat hippocampal CA1 pyramidal cells as described above. However, having the appropriate observation data, most of its tests could easily be adapted to test models of different cell types, even in cases when the experimental protocol is slightly different from the currently implemented ones. The extent to which a test needs to be modified in order to test models of other cell types depends on how much the behavior of the new cell type differs from the behavior of rat CA1 pyramidal cells, and to what extent the protocol of the experiment differs from the ones we used as the bases of comparison in the current study.

## Supporting information

**S1 Appendix. Example of running the Somatic Features Test of HippoUnit using a Jupyter notebook.**
(DOCX)

## Acknowledgments

We thank Judit Makara and her group in the Laboratory of Neuronal Signaling, Institute of Experimental Medicine, Hungary for the patch clamp recording data used in this study. We also thank Luca Tar, a member of our group, for her help in testing the validation tests, and in literature review, and for useful discussions. We also would like to thank Michael Gevaert (Blue Brain Project) for providing the script that finds the apical point, and that were further developed for classifying apical sections.

## Author Contributions

**Conceptualization:** Sára Sáray, Christian A. Rössert, Szabolcs Káli.

**Formal analysis:** Sára Sáray.

**Funding acquisition:** Sára Sáray, Eilif Muller, Tamás F. Freund, Szabolcs Káli.

**Investigation:** Sára Sáray, Szabolcs Káli.

**Methodology:** Sára Sáray, Shailesh Appukuttan, Armando Romani, Andrew P. Davison.

**Resources:** Rosanna Migliore, Paola Vitale.

**Software:** Sára Sáray, Shailesh Appukuttan, Carmen A. Lupascu, Luca L. Bologna, Werner Van Geit.

**Supervision:** Christian A. Rössert, Eilif Muller, Tamás F. Freund, Szabolcs Káli.

**Visualization:** Sára Sáray.

**Writing – original draft:** Sára Sáray, Szabolcs Káli.

**Writing – review & editing:** Sára Sáray, Christian A. Rössert, Shailesh Appukuttan, Rosanna Migliore, Paola Vitale, Carmen A. Lupascu, Luca L. Bologna, Werner Van Geit, Armando Romani, Andrew P. Davison, Eilif Muller, Tamás F. Freund, Szabolcs Káli.

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
