## [Decision Letter · Decision Letter 0]

27 Jul 2020

Dear Ms. Sáray,

Thank you very much for submitting your manuscript "Systematic comparison and automated validation of detailed models of hippocampal neurons" for consideration at PLOS Computational Biology.

Congratulations on an interesting and provocative paper which presents some great technology.

I tend to agree with Reviewer 2 that this might be better presented as a methods paper, with submission of another paper to discuss the contentious consensus issues. It is unclear to me what kind of consensus will be practical to develop at this time: **a single consensus model **for a cell time?; **10 or 1000** consensus models?; or perhaps none possible at this time due to the lack of dendritic data -- very little is known about dendritic voltage patterns -- something that would ideally be provided by a model to be widely used by the community.

Perhaps a perspective paper could be written in a point-counterpoint form; we could ask Rev 2 if he'd be interested in providing the counterpoint?

As with all papers reviewed by the journal, your manuscript was reviewed by members of the editorial board and by several independent reviewers. In light of the reviews (below this email), we would like to invite the resubmission of a significantly-revised version that takes into account the reviewers' comments.

We cannot make any decision about publication until we have seen the revised manuscript and your response to the reviewers' comments. Your revised manuscript is also likely to be sent to reviewers for further evaluation.

Sincerely,

William W Lytton, MD

Guest Editor

PLOS Computational Biology

Kim Blackwell

Deputy Editor

PLOS Computational Biology

Reviewer's Responses to Questions

**Comments to the Authors:**

Reviewer #1: General Comments

* The authors have developed a very interesting, useful tool which promises to

greatly improve the development of biophysically detailed neuron models, and I

hope that tool will go on to be widely used in the field. The data presented

here provide a good demonstration of capabilities of HippoUnit and how it can

be used to improve model development.

* While the authors convincingly demonstrate the utility of their tool, the

manuscript says little in the way of what it can teach us. Though the authors

acknowledge there may be no single model which performs arbitrarily well on all

tests, are any of the tests or optimization methods used to create these models

more or less important for making them more generalizable? For instance, is

their any clear reason the model from Katz et al. 2009 seems to do reasonably

well on most tests, while others have more varied performance?

* A general comment on the figures: They include a lot of information fairly

compactly, and that is commendable, but they are almost impossible to decipher

unless they are viewed in tandem with the main text. Because I think is a tool

that should be adopted by the field, and since many readers tend to skim the

figures for something of interest before reading the manuscript in toto, it

would greatly improve the manuscript and the chances of HippoUnit's wider

adoption if the information in the figures were made more self-evident (more

detailed captions, subfigure labels, more clearly labeled axes, etc.).

* Can you say what it is about the Bianchi model that makes it behave correctly

in the depolarization block test or how the other models go about producing

behavior that "tricks" it?

* It would be very informative to provide some explanation of why the tests

applied here were chosen (Somatic Features, PSP Attenuation, etc.), as well

as what tests could have been included and why they were not.

Specific Comments

* Fig 1 - I'm assuming that the orange/purple coloring of the dendrites is

meant to be basal/apical respectively, but please specify in figure legend.

* Fig 2 - Would it be possible to place scale bars for the morphologies? Or

if they are all on the same scale, please say so in the figure legend.

* Fig 3

** What are the solid lines?

** This is a fairly minimalist caption for such a dense figure. Maybe label

each panel based on what the protocol for finding each of these features.

** The y-axis labels are a bit cryptic, need to dig through text to find them.

** In the legend, what is the difference between 's' and 'w' in reference to

the Golding models?

** It's hard to distinguish experimental values from Poirazi model's values -

consider using asterisks instead of circles.

* Fig 5 - It's not immediately clear that the final/raw scores referred to are

the feature error scores, and that a lower score means a better fit to the data.

More consistency in how scores are referred to would improve the clarity of the

manuscript.

* Fig. 11 has similar issues to Fig 3 - minimal caption, not obvious what one

is looking at without having read the text, y-axis labels are not very

informative. Additionally, it would be nice to indicate on the figure where

statistical significance is achieved.

* HippoUnit - How would one about new experimental data or new tests be

integrated into HippoUnit? Is there a tool in the package that walks the user

through this, or do they have to create their own JSON files 'by hand'?

* The information provided in Lines 232-250 may be better suited to a table.

* Lines 607-610 - I'm not sure where in Methods outlines these useful output

* Line 998: "they own" -> "their own"

* Lines 1066-1068: 'With this synaptic weight the first dendritic spike, and

therefore the jump in the “measured EPSP vs expected EPSP” curve to the

supralinear region is for 4 synaptic inputs, instead of 5.' This is very

confusing, I believe the first comma is misplaced. Regardless, please consider

rewording.

* Line 1277: Poirazi et al. 2013 -> 2003

Reviewer #2: The review is uploaded as a PDF attachment.

**Have all data underlying the figures and results presented in the manuscript been provided?**

Reviewer #1: Yes

Reviewer #2: Yes

PLOS authors have the option to publish the peer review history of their article (what does this mean?). If published, this will include your full peer review and any attached files.

Reviewer #1: No

Reviewer #2: No
---

## [Decision Letter · Decision Letter 1]

16 Nov 2020

Dear Ms. Sáray,

Thank you very much for submitting your manuscript "HippoUnit: A software tool for the automated testing and systematic comparison of detailed models of hippocampal neurons based on electrophysiological data" for consideration at PLOS Computational Biology. As with all papers reviewed by the journal, your manuscript was reviewed by members of the editorial board and by several independent reviewers. The reviewers appreciated the attention to an important topic. Based on the reviews, we are likely to accept this manuscript for publication, providing that you modify the manuscript according to the review recommendations.

Sincerely,

William W Lytton, MD

Guest Editor

PLOS Computational Biology

Kim Blackwell

Deputy Editor

PLOS Computational Biology

[LINK]

Reviewer's Responses to Questions

**Comments to the Authors:**

Reviewer #1: I'd like to thank the authors for addressing my comments and those of the other reviewer. They have addressed all of my original concerns. The figures in particular are a large improvement over their previous version. I also believe the work is far better suited to its present format as a Methods paper and recommend its publication.

Reviewer #2: The authors have performed considerable work to reframe this manuscript as a Methods piece, which I think has improved it dramatically, and I applaud them for putting in this effort. As the authors mention, many of my initial comments on the manuscript have been resolved through this change. At this point, I am ready to endorse publication of this manuscript following the consideration of the following comments, which I think should be easily doable especially relative to the first-stage revisions.

1) The largest issue I had with the revised manuscript was confusion regarding the use of patch clamp versus sharp electrode data. The discussion of this at the beginning of the "Somatic Features Test" section (Lines 685-689) seems to make the argument that the patch clamp data is more appropriate in the setting of the explorations performed in the manuscript. However, beginning around Line 800 the sharp electrode data is reintroduced without much explanation, and it's said that both datasets were used. Then, at Line 821 the authors state that initial explorations show that most models actually fit the sharp electrode recordings better, which confuses things further. Only after some "recalculations" (Line 851) is it concluded that there is a superior fit to the patch clamp data.

Personally, I feel that this needlessly confounds the overall story of the "Somatic Features Test" section, and I'm not sure that the sharp electrode recordings add something "necessary" to the piece (although I acknowledge that, given my confusion interpreting this portion of the manuscript, I may have overlooked other parts of the paper where the sharp electrode data is more necessary). I would ask the authors to review this aspect of the work in detail, and ask themselves three questions: first, whether both data sets are necessary for the totality of the manuscript; second, whether both data sets are necessary in the "Somatic Features Test" section specifically; and third, depending on the conclusions reached for the first two questions, how the necessity of both data sets can be put forth in a more clear and succinct manner.

2) In their response, the authors note that "models in the Golding et al. 2001 paper were not tested as extensively" as others in the manuscript. I wonder if that might unnecessarily confound the comparisons throughout the paper then. Might the paper read more clearly if the Golding models were simply eliminated? This might help to make the paper more digestible overall, as it is already quite voluminous and dense. I leave this choice to the authors, however.

3) I find myself a bit concerned by the author's response to my comment 11, regarding the fact that "some features cannot be evaluated for a given model [because] the model behaves qualitatively differently from the experimental data". While the authors have indeed discussed this in the revised manuscript as mentioned, I believe that this merits a much more detailed discussion in light of the quoted comment above. I could be misunderstanding the author's response, but while it's true that if a model does not exhibit action potentials under a certain protocol spiking features can't be evaluated, if the experimental data exhibits spiking but the model doesn't, this is a major issue. Essentially, I find myself more confused by where HippoUnit stands on this issue based on the author's response to my comment than I was previously. While I don't think this is a "dealbreaker" by any means, I would appreciate at minimum for this to be expounded upon in a response, and ideally for there to be some furtherdetailed discussion of these issues in the text.

4) A couple times in the text (for instance, on line 42) the authors make statements along the lines of "each of these models provides a good match to experimental results in some domains but not in others". This is a somewhat obvious assertion, and I do not believe the authors intend it to be the "crux" of the argument of their paper, but the way it is seemingly highlighted places more weight on this quite intuitive assertion than I think the authors intended. I think such a statement is distracting and can be removed (especially since the authors will "show" this to be the case via their use of HippoUnit, and thus don't need to "tell" us explicitly). Indeed, if a would-be reader were to conclude that this were the overall argument of the paper, they might be unintentionally turned off.

5) The authors mention that the cells of interest are rodent neurons somewhat inconsistently. Especially considering the burgeoning literature on neural modeling in humans and non-human primates, I would ask that the authors consistently refer to "rodent hippocampal CA1 pyramidal cells" to avoid inadvertent confusion.

6) There are typos on Line 82 (a space is needed between 131 and different), potentially on Line 222 (in the mention of the patch clamp dataset), and on Line 1219 (interneurons, not interneuron, as first word).

7) I would eliminate the phrase "common, standardized criteria" on Line 94, which I believe veers too close to the "philosophical" arguments that have been side-stepped by the new Methods format.

8) While I appreciate the inclusion of the first paragraph in the "Interpreting the results of HippoUnit" section, which goes to some of my initial concerns, I feel like this section (along with perhaps the concluding sentence beginning on Line 919 of the "Depolarization Block Test" section) may read to some like "counter arguments" to the paper itself. It is somewhat jarring to read about the importance of the creation of HippoUnit, who's primary output is scores on validation tests, only to later be told that "a high final score on a validation test using a particular experimental dataset does not mean that the model is not good enough or cannot be useful for a variety of purposes". Ideally, I would like to see this "tone" included consistently throughout the paper, but I recognize that may be a manifestation of my own "philosophical" bias. At this point I do not think that this issue is a "dealbreaker" when it comes to publication, but I would ask that the authors spend some time thinking about how these critical caveats might be made to seem less jarring when they're brought up... this could perhaps be accomplished just by some increased foreshadowing at appropriate points in the text.

While this may seem like a long list, I emphasize that I am enthusiastically endorsing the publication of this manuscript with these comments considered, and I feel like these are relatively minor issues that likely arose given the significant rewrites that the authors performed. It is inevitable that some new issues would arise following such a significant rewrite, and once again I applaud the authors for putting forth the time and effort to do so. With these issues addressed, I feel this will be a more complete Methods manuscript and a strong publication in PLoS Computational Biology.

**Have all data underlying the figures and results presented in the manuscript been provided?**

Reviewer #1: Yes

Reviewer #2: Yes

PLOS authors have the option to publish the peer review history of their article (what does this mean?). If published, this will include your full peer review and any attached files.

Reviewer #1: No

Reviewer #2: No
---

## [Decision Letter · Decision Letter 2]

24 Dec 2020

Dear Ms. Sáray,

We are pleased to inform you that your manuscript 'HippoUnit: A software tool for the automated testing and systematic comparison of detailed models of hippocampal neurons based on electrophysiological data' has been provisionally accepted for publication in PLOS Computational Biology.

Best regards,

William W Lytton, MD

Guest Editor

PLOS Computational Biology

Kim Blackwell

Deputy Editor

PLOS Computational Biology

Reviewer's Responses to Questions

**Comments to the Authors:**

Reviewer #2: I once again applaud the authors for their engagement in this revision process and willingness to meaningfully respond to my queries and make corresponding changes in the manuscript. The most recent round of responses and edits adequately addresses all of my previous concerns, and as such I now fully endorse this manuscript's publication.

**Have all data underlying the figures and results presented in the manuscript been provided?**

Reviewer #2: Yes

PLOS authors have the option to publish the peer review history of their article (what does this mean?). If published, this will include your full peer review and any attached files.

Reviewer #2: No

---

## [Editor Report · Acceptance letter]

22 Jan 2021

PCOMPBIOL-D-20-01121R2 

HippoUnit: A software tool for the automated testing and systematic comparison of detailed models of hippocampal neurons based on electrophysiological data

Dear Dr Sáray,

I am pleased to inform you that your manuscript has been formally accepted for publication in PLOS Computational Biology. Your manuscript is now with our production department and you will be notified of the publication date in due course.

With kind regards,

Jutka Oroszlan
